

# Measurement report: Violent biomass burning and volcanic eruptions: a new period of elevated stratospheric aerosol over Central Europe (2017 to 2023) in a long series of observations

Thomas Trickl[1], Hannes Vogelmann[1], Michael D. Fromm[2], Horst Jäger[1] and Matthias Perfahl[1]

[1]Karlsruher Institut für Technologie, Institut für Meteorologie und Klimaforschung (IMK-IFU), Kreuzeckbahnstr. 19, D-82467 Garmisch-Partenkirchen, Germany

[2]Naval Research Laboratory, 4555 Overlook Avenue, Washington, D.C., U.S.A.

*Correspondence to:* Dr. Thomas Trickl, thomas@trickl.de, Tel. +49-8821-50283

**Abstract.** The highlight of the meanwhile 50 years of lidar-based aerosol profiling at Garmisch-Partenkirchen has been the measurements of stratospheric aerosol since 1976. After a technical breakdown in 2016, they have been continued with a new, much more powerful system in a vertical range up to almost 50 km a.s.l. that allowed to observe very weak volcanic aerosol up to almost 40 km. The observations since 2017 are characterized by a number of spectacular events, such as the Raikoke volcanic plume equalling in integrated backscatter coefficient that of Mt. St. Helens in 1981 and severe smoke from several big fires in North America and Siberia with backscatter coefficients up to the maximum values after the Pinatubo eruption. The smoke from the violent 2017 fires in British Columbia gradually reached more than 20 km a.s.l., unprecedented in our observations. The sudden increase in frequency of such strong events is difficult to understand. Finally, the plume of the spectacular underwater eruption on the Tonga islands in the southern Pacific in January 2022 was detected between 20 and 25 km.

*Key words:* Lidar, system upgrade, stratospheric aerosol

## 1 Introduction

In view of its impact on the radiation budget and air chemistry the stratospheric aerosol layer has been monitored since 1972 with balloon- and satellite-borne sensors as well as with lidar (Deshler 2006; 2008; Kremser et al. 2016; Vernier et al., 2016; Bingen et al., 2017; Thomason et al., 2018; 2021). Ground-based lidar with its good vertical resolution became an important tool almost right from the beginning (McCormick et al., 1978; Simonich and Clemesha, 1997). A number of stratospheric aerosol sounding stations provided routine long-term measurements at different latitudes (e.g., Osborn et al., 1995; Jäger, 2005; Deshler et al., 2006; 2008; Trickl et al., 2013; Khaykin et al., 2017; 2018a, b; Zuev et al, 2017; 2019; Chouza et al., 2020) and since the 1990s in part adopted by the Network for the Detection of Stratospheric Change (NDSC, now: NDACC, Network for the Detection of Atmospheric Composition Change).

The observations have yielded evidence of the mainly volcanic nature of the stratospheric aerosol. Long-lasting aerosol loading is only expected for a significant penetration of a volcanic plume into the stratosphere (Deshler, 2008). Secondary sources are strong injections from biomass burning (e.g., Fromm and Servranckx, 2003; Fromm et al., 2000; 2008a, b; 2010; 2019; 2022; de Laat et al., 2012; Khaykin et al., 2020; Lestrelin et al., 2021,



Peterson et al., 2021), likely to be more important in a warmer climate, or emissions by air traffic. Also a potential influence of the growing Asian $SO_2$ emissions from coal burning has been discussed (Hofmann et al., 2009; Vernier et al., 2015). In addition, desert dust from Africa and Asia has been observed in the lower

stratosphere (Trickl et al., 2013; Murphy et al., 2021).

The mid-latitude stratospheric aerosol level after a significant particle injection is subject to decay. Time series yield a decay time of about five years for tropical eruptions such as El Chichon (1982) and Pinatubo (1991). This is explained by an atmospheric updraft creating a tropical stratospheric reservoir layer. Poleward transport of aerosol from this reservoir in the Brewer-Dobson circulation (Trepte and Hitchman, 1992; Butchart et al., 2006;

Butchart, 2014) leads to filling the mid-latitude losses by downward transport for many years. Aerosol removal is also due to dilution (Fromm et al., 2008b) and, in the mid-latitudes, by processes like tropopause folding (Holton et al., 1995; Stohl et al., 2003). In fact, aerosol has been observed in stratospheric air intrusions into the troposphere after pronounced eruptions (e.g., Browell et al., 1987; Trickl et al., 2016). Total decay times for mid-latitude eruptions or fires are of the order of one year or less.

North American and Siberian fires can yield very strong contributions over Europe. Pronounced Canadian events have been frequently observed in the troposphere (e.g., Forster et al., 2001; Müller et al., 2005; Petzold et al., 2007; Ancellet et al., 2016; Trickl et al., 2015; Markowicz et al., 2016). Also in the stratosphere, dense Canadian smoke plumes have been observed with growing occurrence (e.g., Fromm et al., 2010, 2020; 2022; and other papers cited above). The most spectacular event was that of the wild fires in British Columbia starting in August

2017. Peterson et al. (2018) and Fromm et al. (2021) report that the mass of smoke aerosol particles injected into the lower stratosphere from five near-simultaneous intense pyro-cumulonimbus (pyroCb) events occurring in western North America on 12 August 2017 was comparable to that of a moderate volcanic eruption, and one order of magnitude larger than previous benchmarks for extreme pyroCb activity.

These plumes were registered and followed in time at many stations within EARLINET (Khaykin et al., 2018;

Ansmann et al., 2018; Baars et al., 2018; EARLINET: European Aerosol Lidar Network, Bösenberg et al., 2003). As we will discuss in this paper (Sect. 3) the smoke gradually rose to more than 20 km above the Northern Alps. Ansmann et al. (2018) determined an extreme aerosol optical thickness (AOT) close to 1.0 at 532 nm in this layer that crossed central Europe at a height of 3 to 17 km on 21 to 22 August 2017. They concluded from measurements at three stations (Leipzig, Hohenpeißenberg (both Germany) and Kosetice (Czech

Republic)) that the stratospheric light-extinction coefficients observed at a height of 14 to 16 km, were up to twenty times higher than the maximum extinction coefficients reached after the Mt. Pinatubo eruption in June 1991 (Ansmann et al., 1997; Jäger, 2005).

This event was just the first of several strong fires that yielded significant aerosol loading of the stratosphere in recent years. In this paper, we discuss the related measurements in Garmisch-Partenkirchen (Germany). The period

since 2017 has been one of the most interesting segments in the long-term stratospheric lidar sounding series that now covers a total of 47 years (Jäger, 2005; Trickl et al., 2013). We first outline the history of the three lidar systems so far used (Sect. 2.1) and describe the most important properties of the demanding data-evaluation procedure in order to underline the quality of the data (Sect. 2.2). In Sect. 3.1 we present the Garmisch-Partenkirchen series of the stratospheric integrated aerosol backscatter coefficient presented that extends from October 1976 to

(currently) May 2023, including a few gaps due to technical issues. In Sects. 3.2 and 3.3, we analyse the rather



eventful period since 2017 with five aerosol peaks in the lower stratosphere from the 2017 fires in British Columbia, the Raikoke volcanic eruption, the Colorado fires in autumn 2020, violent fires in British Columbia in 2021, and the highly explosive Tonga eruption in 2022. The analysis benefits for the first time in our series from transport modelling over almost two weeks combined with satellite data, as well as on information in the source

region and information available on the aerosol bursts themselves.

## 2 Methods

### 2.1 System history

The stratospheric lidar measurements were made until January 2016 with two lidar systems at IMK-IFU (until 2001 IFU, i.e., **I**nstitut **f**ür Atmosphärische **U**mweltforschung of the Fraunhofer Society; 47º 28′ 37″ N, 11º 3′

52″ E, 730 m a.s.l.). In the following, lidar operation was resumed with a new system at the nearby high-altitude station Schneefernerhaus (Umweltforschungsstation Schneefernerhaus, UFS, 47° 25′ 00″ N, 10° 58′ 46″ E, 2675 m a.s.l.) on the south side of Mt. Zugspitze (2962 m a.s.l.), about 9 km to the south-west of IMK-IFU.

#### *Ruby lidar*

The first system was delivered in 1973 by Impulsphysik G.m.b.H., based on a ruby laser, and, in addition to a

large number of routine and campaign-type tropospheric measurements (e.g., Reiter and Carnuth, 1975; Jäger et al., 1988). After adding a photon-counting system the lidar was almost continually used for night-time measurements of stratospheric aerosol since autumn 1976 (Reiter et al., 1979; Jäger, 2005).

#### *Lidar container*

After the ruby laser quit operation in 1990 the lidar was rebuilt in a container as a transportable, spatially

scanning system with a Nd:YAG laser (Quanta Ray, GCR 4, 10 Hz repetition rate, about 700 mJ per pulse at 532 nm), starting in 1991 for additional investigation of contrails (Freudenthaler, 2000; Freudenthaler et al., 1994; 1995). The 0.52-m Cassegrain receiver of the 1973 system was retained. The laser was delivered early enough to resume the measurements just before the Pinatubo eruption. The lidar was used for both free-tropospheric (e.g., Jäger et al., 2006; Forster et al., 2001; Trickl et al., 2003; 2011) and stratospheric (Jäger, 2005) measurements.

The vertical bins of this 300-MHz multichannel scaler (FAST ComTec) were set to 75 m. Four subsequent measurements were made without attenuation and with three different attenuators, the strongest one being used for the near-field detection. A high-speed chopper was set to cut off the strongest part of the signal. For each attenuation step a different chopper delay was applied (minimum distance achieved: 1.3 km).

This second lidar system contributed to both NDACC (Network for the Detection of Atmospheric Composition

Change; www) and EARLINET. The data in the data bases are not smoothed which in the case of strong extinction leads to noise spikes in the lowest data segments up to the tropopause region where the photon-counting signals were attenuated for minimizing counting dead-time (pulse-overlap) effects. With this lidar system rather small aerosol structures exceeding roughly 2 % of the Rayleigh return at 532 nm (that corresponds to a visual range of more than 400 km above 3 km) could be resolved within the free troposphere and lower



stratosphere. The aerosol backscatter coefficients could be calculated with a relative uncertainty of 10 to 20 % under optimum conditions.

This lidar container was used to extend the stratospheric aerosol series (Jäger, 2005; Trickl et al., 2013) until 2016. The end was caused by a degradation of the container and components, however eventually fatal problems in the data transfer from the counting system to the computer after a measurement. These problems are reflected

by the diminishing number of stored data in 2014 and 2015 and led to abandoning the system in early 2016.

***UFS lidar (2675 m a.s.l.)***

The new lidar at UFS is integrated into the water-vapour differential-absorption lidar (DIAL; Vogelmann and Trickl, 2008) by sharing its 0.65-m-diameter Newtonian receiver (providing a 56-% gain in area) and its polychromator box. On 29 September 2017 a Spitlight DPSS frequency-doubled injection-seeded Nd:YAG laser

from Innolas (wavelength 532.24 ± 0.02 nm) replaced the pump laser of the DIAL. Thus, the repetition rate could be increased from 20 Hz to 100 Hz, the second-harmonic pulse energy being 140 mJ instead of 200 mJ. The number of laser shots in a single measurement was gradually increased to 100000 (16.7 min; Table 1).

The operation of the system at this elevated site offers the benefit of much clearer average conditions frequently outside the Alpine boundary layer (e.g., Carnuth and Trickl, 2000; Carnuth et al., 2002), including cloud-free

conditions during night-time. In addition, despite shortening the distance to the stratosphere by just 1945 m a considerable gain in signal is obtained. This is explained by the extreme near-field drop of the backscatter signal with the distance and the requirement to select the same setting of the maximum detector output voltage at both locations (typically 70 mV into 50 Ω for the detector type used, see below). A simulation shows that even at 25 km a.s.l. the gain in signal is still a factor of 1.5. This factor, together with the larger receiver, helps to avoid a lot

of expensive additional laser photons.

The polychromator of the DIAL is used as described by Vogelmann and Trickl (2008). Near-field and far-field signals are separated by a beam splitter. The far-field channel contains a blade placed in a focal point that cuts off the near-field return. Residual background radiation (e.g., scattered light from local sources) is strongly reduced by an interference filter with 0.5 nm full width at half maximum (Barr Associates). In the near-field

channel the very strong return is attenuated by two decades by a neutral-density filter (Andover).

The electronics used share the highly linear approach of the lidar systems of IMK-IFU (Trickl et al., 2020a; Klanner et al., 2021). The high linearity of the data is ensured by Hamamatsu R7400U-03 photomultiplier tubes (with actively stabilized socket and a high-speed discriminator junction from Romanski Sensors, RSV), Licel transient digitizers (12 bits, 20 MHz, equipped with ground-free input) and a FastComtec MCS6A 5-GHz

photon-counting system. The data are processed at 7.5-m height intervals.

One great advantage of the new data acquisition is that it is no longer exclusively based on single-photon counting as until 2015 which had required strong attenuation of the signals in the case of near-field detection in order to avoid the photon-pulse overlap issues. Without attenuation the relative contribution of the near-field noise is greatly reduced. The analogue signal for the near-field and far-field detectors is fully linear up to

distances r of more than 15 km and more than 30 km, respectively (after a tiny exponential correction, Trickl et al., 2020a). Due to the narrow spectral filtering the noise from the solar background is sufficiently reduced to



allow daytime measurements with the analogue channels up to 30 km. The best performance of the analogue channels is achieved if the peak signal is kept below 70 mV.

The photon-counting data are useful without smoothing to more than 50 km a.s.l. The PMT tests over one hour
(Klanner et al., 2021) have demonstrated the absence of dark counts (thermal emission from the photocathode). The night-time lidar background is not fully zero (about 50 counts) which may be improved.

Another great advantage of this system over the old ones is that it can be operated under remote control, in particular benefitting from the corresponding features of the new laser. During a field campaign in the United States in summer 2018 the measurements were started from the other side of the Atlantic Ocean.

We also plan to add a 355-nm channel, a depolarization channel as well as a 532-nm high-spectral-resolution channel for extinction measurements during periods of strong stratospheric aerosol loading, as a contribution to the European ACTRIS (Aerosol, Clouds and Trace Gases Research Infrastructure) network.

**2.2 Data evaluation**

The quality of stratospheric aerosol backscatter coefficients critically depends on the procedures applied.
Because of the ongoing discussions within NDACC we outline in the following the most important properties of the approach chosen. The careful procedure, just very briefly sketched by Jäger (2005), has been refined, motivated by the improved signal-to-noise ratio of the new system.

For the measurements until 2011 (Jäger, 2005; Trickl et al., 2013) an iterative approach for calculating the aerosol backscatter coefficients was chosen. Afterwards, an extended-Klett (Klett, 1985) program originally
developed and very successfully quality assured for aerosol retrievals within EARLINET. The sign error in Eq. 20 of Klett (1985) is corrected, yielding Eq. 2 of Eisele and Trickl (2005; see also Speidel and Vogelmann, 2023). The Klett downward inversion typically starts at a distance of 45 km (47.675 km a.s.l.). This program uses an approach for the extinction-to-backscatter ratio (lidar ratio) with up to ten layers. Since 2012, a lidar ratio of 50 sr has been applied in the troposphere, 45 sr in the stratosphere. The latter value is valid approximately
within ± 7 sr for periods outside the extreme eruptions of El Chichon and Pinatubo (Jäger and Deshler, 2003). Because of the mostly low extinction of stratospheric aerosol the choice of the lidar ratio is normally not critical. In the presence of cirrus clouds or particularly strong aerosol peaks at least one additional layer is introduced whenever a calibration of the aerosol backscatter coefficients is possible below the clouds (Eisele and Trickl, 2005). In cirrus layers typical values of the lidar ratio of 10 sr to 30 sr are retrieved, the higher values most likely
corresponding to cases of non-persistent clouds during the measurement period.

A key issue of the retrieval is an accurate calculation of the Rayleigh backscatter coefficients (see Appendix). This requires the calculation of the atmospheric air density from sufficiently accurate meteorological data. The molecular return is simulated by calculating the atmospheric density from the routine radiosonde ascents at Oberschleißheim ("Munich" radiosonde, station number 10868, 101 km roughly to the north;
http://weather.uwyo.edu/upperair/sounding.html) and, above the maximum altitude of the sondes, by using NCEP (National Centers for Environmental Prediction) meteorological data up to more than 50 km, daily interpolated for our station for 12 UTC (13 CET; meanwhile available at: https://www-air.larc.nasa.gov/misions/ndacc/data.html#). All sonde and NCEP altitudes are converted from geopotential to absolute units.



The new approach accounts for Raman scattering (see Appendix). In the near-field channel the complete Raman
band is detected. The interference filter in the far-field channel cuts off most of the S- and O-branch
contributions of oxygen and nitrogen.

As an example for the high data quality we show in Figure 1 the results of the retrieval for a measurement on 7
January 2021 between 18:18 and 18:35 CET (Central European Time = UTC + 1 h). The Klett solutions for the
three detection channels are displayed. The agreement of the curves in regions of overlap is excellent, after very
small exponential corrections of the analogue signals (Sect. 2.1) that are optimized by comparison with the
photon-counting signal in aerosol-free altitude ranges. At large distances r the noise of the values from the
photon-counting data is significantly lower than that of the analogue data and no artificial structure is seen. The
counting noise level in the raw data descends with altitude. Due to the multiplication of the signals with $r^2$ during
the Klett inversion an almost constant noise level is reached.

The calibration of the backscatter coefficients is additionally controlled by the requirement that the backscatter
coefficients must stay positive. Here, it is beneficial that layers without aerosol quite frequently occur in the
upper troposphere.

The backscatter signals have been corrected for the light absorption by ozone in the stratosphere cross section for
both the ruby wavelength (before 1990) and the wavelength of the frequency-doubled Nd:YAG laser (Brion et
al., 1998; http://igaco-o3.fmi.fi/ACSO/cross_sections.html). The values for $\lambda_{air}$ = 532.092 nm are almost
temperature independent, ranging between $2.812 \times 10^{-25}$ m$^2$ (295 K) and $2.805 \times 10^{-25}$ m$^2$ (218 K). Climatological,
seasonally varying ozone profiles have been taken, kindly provided by the nearby Meteorological Observatory
Hohenpeißenberg of the German Weather Service (MOHp). Figure 2 shows that these corrections are by no
means negligible in the altitude range of maximum ozone concentrations.

An accurate determination of the tropopause altitude is crucial for the accurate calculation of the integrated
backscatter coefficient in the stratosphere. It is normally extracted from the temperature data from the Munich
radiosonde. Both the values for the WMO criterion (WMO, 1986) and the temperature minimum are calculated.
These values have been compared since 2012 with a number of ancillary data and modified if necessary (in rare
cases). The validation and modification is derived from the ozone rise provided by the ozone differential-
absorption lidar at IMK-IFU (Trickl et al., 2020a) whenever available, the drop of relative humidity in the sonde
data and the upper edge of cirrus clouds (if present). Also the aerosol distribution in the tropopause region (e.g.,
cirrus clouds) has been used for refinements in unclear situations.

Because of the strong drop in signal the near-field raw data were smoothed with a linearly growing interval,
using the Blackman-type numerical filter of Trickl et al. (2020a). At r = 10 km the interval size reaches ±13 bins
(±100 m), corresponding to a vertical resolution of roughly 40 m in the VDI definition and roughly 70 m as
defined by the full width at half maximum of the response to a delta function (Trickl et al., 2020a).

A discussion of the uncertainties is given in the Appendix.

## 3 Results

### 3.1 Series of the integrated backscatter coefficient (1976 – 2023)



220 Figure 3 shows the updated version of the time series of the stratospheric integrated backscatter coefficient from October 1976 to May 2023. The integration starts at 1 km above the tropopause in order to reduce the influence of contributions of mixed tropospheric and stratospheric character. Since we cannot easily repeat the evaluation of the old data with all the microphysical details we continued our tradition and converted the values from 532.24 nm to the ruby-laser wavelength of 694.3 nm (Jäger and Deshler, 2002; 2003).

225 In 2014 and 2015 the number of the measurements that could be stored in the computer strongly diminished due to data transfer issues, and the old lidar system was abandoned after a final measurement on 29 January 2016. Typical scattering ratios were about 1.05, i.e., rather low, but higher than the very small pre-2006 background. With the new system the routine sounding was resumed in 2017, after one test measurement on 17 March 2016. Until the end of the series several pronounced peaks are seen. The decay in 2022 led to values next to the 1979

230 level, before a new phase of elevated aerosol prevented a return to a more pronounced background phase. In Table 1 we list some of the conditions of the measurements since 2017.

There is an obvious winter-summer modulation, visible in particular during the low-background period. This is mostly due to the changing tropopause. For example, the peak in January 2019 is mainly caused by tropopause altitudes of about 10 km, whereas they were of the order of 12 km in February. The seasonal cycle is somewhat

235 obscured by occasional plumes just above the tropopause.

Between early 2014 and August 2017 there were just three major eruptions, all in the tropics (https://volcano.si.edu/; Massie, 2016; more information on individual cases: https://volcano.si.edu/index.cfm) and, thus, not so important for observations at our latitude, but can contribute to the background with a delay of the order of half a year (Jäger, 2005). The Kelut plume (Java, volcanic explosivity index (Newhall and Self,

240 1982) VEI = 4) reached as much as 17 km on 13 February 2014, and Manam (New Guinea) 19.8 km in a possibly brief explosion on 31 July 2015. The eruption of Cotopaxi is special since this volcano in Equador is the highest active volcano on this planet (5897 m). The material reached 17.9 km a.s.l. on 15 August 2015, not far from the tropical tropopause.

Between 2017 and February 2023 three eruptions may have influenced our series. In 2019 the volcanoes Raikoke

245 (Kuril Islands) and Ulawun (New Guinea) spewed material into the stratosphere. On 15 January 2022 a particularly explosive eruption was reported on the Tonga islands that reached about 58 km (Proud et al., 2022; Taha et al., 2022). Due to its occurrence in the southern hemisphere related particles were detected above Garmisch-Partenkirchen not before October, except for a single, accidental observation on 29 June 2022.

The pronounced peaks of the integrated backscatter coefficient registered with the new system since 2017 will be

250 discussed in the following. Most importantly, in addition to the volcanic eruptions, a number of exceptionally violent fires led to significant rises in stratospheric aerosol. These fire events make this period particularly interesting, with peak backscatter coefficients and peak altitudes unprecedented for smoke in our series. In contrast to earlier years a much better analysis of the sources of stratospheric aerosol has become possible by a combination of extended transport modelling and satellite data.

255 **3.2 Sudden increase of the occurrence of fire plumes deeply penetrating into the stratosphere**

Until July 2017 volcanic eruptions were the main source of stratospheric aerosol detected above our site. An exception was the remarkable aerosol loading following a fire in Québec in June 1991 just before the arrival of




the Pinatubo plume (Fig. 3) that reached roughly 17 km above Garmisch-Partenkirchen (Carnuth et al., 2002; Fromm et al., 2010). After the big Chisholm fire in 2001 the IFU lidar detected aerosol to more than 6 km above

the tropopause (Fromm et al., 2008, Fig. 3) for a short period of time. However, in the measurements in recent years penetration into the stratosphere to more than 20 km has been observed.

Although an increase in the occurrence of strong fires might be expected in a drier climate (Fig. 2 of Trickl et al., 2013) the sudden rise in the number of cases and their outstanding violence is rather surprising.

### *British Columbia fires 2017*

The first, particularly spectacular signature after resuming the measurements at UFS in 2017 was identified as the result of the pyroCbs in British Columbia (B.C.) on 11 and 12 August 2017 that acted like a volcanic eruption (Peterson et al., 2018; Fromm et al., 2021). In general, the B.C. fire season during that year lasted several months, starting on 6 July. The tropospheric smoke plumes were repeatedly detected with the 313-nm channel of our ozone DIAL.

Our first observation of the B.C. plume with the UFS aerosol lidar took place on 25 August (Fig. 4), after a 17-day period without measurements. The daytime measurements (8 CET to 12 CET) with analogue data acquisition, apart from a few smaller peaks, showed a giant aerosol spike as high as 16 to 17 km a.s.l. The 532-nm backscatter coefficients ranged between $1.3 \times 10^{-6}$ m$^{-1}$ sr$^{-1}$ and $1.9 \times 10^{-6}$ m$^{-1}$ sr$^{-1}$, corresponding to scattering ratios (ratio $\beta_R + \beta_P)/\beta_R$ of the backscatter coefficients) between 7.2 and 10.3. This value is exceptionally high for

the stratosphere. For comparison, the Pinatubo maximum scattering ratio above our site in early 1992 was about 10 at 21 km. As mentioned in the introduction, even higher stratospheric aerosol loading caused by the B.C. fires was reported by Ansmann et al. (2018) farther to the north, a few days earlier.

The first burst of pyroCbs was reported by Fromm et al. (2021) for 12 August, at about 23:00 UTC, the last one at about 6:45 UTC on the following day. Several times altitudes between 13 km and 13.7 km were recorded by

the Prince George radar.

Forward simulations with the HYSPLIT model (https://www.ready.noaa.gov/ HYSPLIT_traj.php; Draxler and Hess, 1998; Stein et al., 2015) initialized over the pyroCb area at that time showed passage above the Bavarian Alps on 20 and 22 August, in agreement with the observations by Ansmann et al. (2018 carried out before our first available measurement day (25 August, see Fig. 4). The forward trajectories do not show the rise to more than 16

km that is documented in Fig. 4. One cannot exclude a thermally induced rise (de Laat et al., 2012; Khaykin et al., 2018; Lestrelin et al., 2021). The rise of the plume was verified by the "curtains" of the space lidar CALIOP (Cloud-Aerosol Lidar with Orthogonal Polarization; https://www-calipso.larc.nasa.gov/ products/lidar/ browse_images/products/). Fromm et al. (2021) shows a rise from 13 km over the Canadian western arctic sea to more than 15 km over the northern part of the Hudson Bay. Lestrelin et al. (2021) followed the CALIOP images

to Europe and found splitting of the plume into three parts and a rise to higher altitudes.

An approximate source receptor relationship for 25 August was established by 315-h HYSPLIT ensemble backward trajectories (reanalysis mode), run for start altitudes above Garmisch-Partenkirchen between 15300 m and 17200 m a.s.l. at intervals 50 m. In this altitude range two principal branches are seen, one showing air



passing over the United States (southern branch, SB), one reaching arctic Canada (northern branch, NB). Below
15850 m the SB is almost exclusive. Above this altitude the NB becomes increasingly important.

In numerous transport studies (e.g., Trickl et al., 2013; 2015; 2020b) during the past decade we found that the
HYSPLIT trajectories explain our observations better in the reanalysis mode (using National Center for
Environmental Prediction Reanalysis data). In this case and a case presented further below, the GDAS (Global
Data Assimilation System) mode performs significantly better. In Fig. 5 we show the GDAS result for the
altitude of 16860 m which yields the best proximity to the source region. The NB trajectories pass over the
Arctic regions where the plume was located with the CALIOP images and almost perfectly hit the pyroCb source
region within less than half a day of the most active period, on 12 August after 12 UTC. Given the uncertainty of
trajectory calculations, the considerable spreading of the trajectories towards the west and the complex
meteorology associated with the hot pyroCbs (Lestrelin et al., 2021) this result is highly satisfactory. Most
importantly, we do not know about any other similarly strong aerosol source for that period.

Fromm et al. (2021) show in their Figs. 3 and 9 the propagation of the densest part of the smoke close to the
western end of the Great Slave Lake. In Fig 5, the trajectories pass this area farther to the west where less aerosol
is depicted in that figure. This could explain why our peak backscatter coefficient is lower than that published by
Ansmann et al. (2018). The backward trajectories slightly descend towards the source region, not enough to
exclude a thermal rise of the plume.

Between 25 August 2017 and the end of the year a total of 30 measurements were conducted. For the series
image in Fig. 3 just one measurement per day was selected. Frequently daytime measurements took place. Until
spring 2018 just analogue data acquisition was available that is fortunately highly linear after just minor
correction (Sect. 2, Table 1).

On the following measurement day, 29 August (not shown), the big feature at 16.5 km from the first passage
over Central Europe had disappeared. In October another maximum of the integrated backscatter coefficients
was reached, tentatively ascribed to a more dispersed phase of the plume with a higher probability to pass over
Southern Germany. The plume decayed considerably until December 2017 (for two autumn examples see Fig.
4). The layer top rose from 17 km (25 August 2017) to more than 24 km in February 2018. This is slightly above
the highest altitudes reported by Baars et al. (2019) in an overview for the EARLINET stations. The spiky
structure of the backscatter profile gradually became smoother and disappeared in winter 2018 (not shown). The
short decay of less than one year is typical of mid-latitude aerosol plumes in the stratosphere, as can be
concluded from Fig. 3 (e.g., St. Helens, 1980).

Apart from spreading to other latitudes stratospheric aerosol can be diminished by removal from the lowermost
stratosphere in tropopause folds (e.g., Browell et al., 1987; Trickl et al., 2016). Also in autumn 2017 aerosol was
found in stratospheric intrusion layers in our ozone soundings in the valley at IMK-IFU, this time during a
period of aerosol from biomass instead of a volcanic plume in the lower stratosphere. Figure 6 shows three ozone
and 313-nm-aerosol profiles obtained with our ozone DIAL (Trickl et al., 2020a) in the morning of 2 October
2017, before the arrival of clouds (see aerosol spike at 9.45 km) around noon stopped the measurements. The
descending intrusion layer, originating at more than 10 km over Northern Canada tree to four days earlier, is
characterized by ozone mixing ratios up to almost 140 ppb, in agreement with the morning balloon-borne
measurement at Hohenpeißenberg, 38 km to the north from UFS (Trickl et al., 2023). The minimum



Hohenpeißenberg sonde RH is constantly 2 %, most likely a wet bias (Trickl et al., 2014). The aerosol backscatter coefficient in this layer reached $2\times10^{-7}$ sr$^{-1}$ m$^{-1}$ which is moderate in comparison with other cases, in

particular the record-setting $2.35\times10^{-6}$ m$^{-1}$ sr$^{-1}$, observed on 7 September 2009 after the violent eruption of Sarychev (Trickl et al., 2020b). It is interesting to see that the elevated aerosol backscatter coefficients do not fill the entire intrusion layer although the width scales similarly as that of elevated ozone.

### Siberian fires 2019

Parallel to violent volcanic eruptions (Sect. 3.3) extreme and long-lasting wildfires in central and eastern Siberia

were reported in the summer of 2019 (Johnson et al., 2021. Ohneiser et al. (2021) describe an Arctic field campaign in September and October 2019 with an advanced multi-wavelength polarization Raman lidar onboard the German icebreaker Polarstern. The high lidar ratio suggested that observed at stratospheric aerosol at high latitudes were caused by import of fire smoke from Siberia. Also over Central Europe they report for Leipzig and other lidar stations a contribution of these fires. In the absence of strong pyro-convection they conclude that the

biomass-burning particles were limited to altitudes up to 13 km.
It is difficult for us without the planned high-spectral-resolution detection channel to distinguish between the volcanic aerosol and the Siberian smoke. For more information on this period see Sect. 3.3.

### Colorado fires 2020

The measurements in November and December 2020 were characterised by numerous spikes in the backscatter

profiles in the upper troposphere and lower stratosphere (Fig. 7; the number of profiles is too low for a contour plot). The rather narrow structures on a broad background indicate sources just shortly backward in time. We ascribe these narrow aerosol layers to very late fires in Colorado. According to the MODIS (Moderate Resolution Imaging Spectroradiometer on board NASA's Aqua satellite) web site the 2020 Colorado fire season has been devastating and record-breaking. The three largest fires in Colorado history all occurred during this

year (https://modis.gsfc.nasa.gov//gallery/individual.php?db_date=2020-10-27). The two most violent fires occurred in October 2020, very late in the year. The Cameron Peak Fire burned 844 km$^2$ and the East Troublesome Fire burned 779 km$^2$. The Cameron Peak Fire began in August and is the largest fire in state history, the nearby East Troublesome pyroCb ignited on October 14 and explosively grew until 21 October (13.2 km) to capture the number-two title. At least 11 fires continued to be active in the state on October 26.

The plume was traceable around the world with satellite-based instruments such as MODIS, CALIOP, and vertical soundings of the Micropulse Lidar Network (MPLNET, https://mplnet.gsfc.nasa.gov/). The particles from the East Troublesome October 21 pyroCb in Colorado can be seen to flow over the Atlantic Ocean to reach Europe and North Africa on 25 October. Our first lidar measurement after the occurrence of the pyroCb on 28 October (not shown) shows small aerosol peaks above the smooth background between the tropopause and 14

km that could be a first indication of the fire plume. HYSPLIT ensemble backward trajectories initiated in this altitude range show zonal flow from Colorado to the Alps within just three days. The measurements during the following weeks (Fig. 7) show strongly varying structures and were not analysed in detail because of this complexity except for 9 November. The source-receptor relationship for the lidar measurements UFS on 9



November was hardened by the satellite and MPLNET measurements mentioned above, connected via HYSPLIT forward and backward trajectories. HYSPLIT ensemble trajectories initiated at and around 14.5 km a.s.l. above UFS on 9 November show a rather coherent near-zonal flow around the globe passing over North America twice within 315 h.

While the East Troublesome pyroCb is a plausible source for these November stratospheric layers, we cannot rule out contributions from earlier pyroCbs in the USA. For instance, the Creek fire in California developed its

own spectacular pyroCb on 3 September, injecting smoke upward of 16 km (Hu et al., 2022; Lareau et al., 2022).

***Spectacular pyro-cumulonimbus in British Columbia on 30 June 2021***

The measurements on 11 and 21 July 2021 show pronounced stratospheric aerosol signatures that lead to clearly enhanced backscatter coefficients (Fig. 3). We discuss here just the particularly spectacular case of 11 July (Fig. 8). We observed enhanced aerosol structure in the upper troposphere and in the stratosphere up to about 19.5 km.

This upper edge is absent in earlier measurements during this season. There are four pronounced spikes between 13 and 16 km. The big spike at 15.6 km with a remarkable backscatter coefficient of $7.07 \times 10^{-7}$ m$^{-1}$ sr$^{-1}$ is very thin which indicates an event just a few days backward in time. The maximum value is more than half that obtained for the 2017 fires (see above). The 532.2 nm integrated backscatter coefficient rose to $6.59 \times 10^{-4}$ sr$^{-1}$ ($3.16 \times 10^{-4}$ sr$^{-1}$ at 694.3 nm, Fig. 3).

We relate this observation to high pyroCb activity on 30 June, again in British Columbia. There are reports on record-setting temperatures of more than 45º C in that region, a drought, numerous lightning strokes under dry conditions and huge fires. An article of the Washington Post of 1 July 2021 (https://washingtonpost.com/ weather/2021/07/01/wildfires-british-columbia-lytton-heat/ gives a good overview, citing several scientists, and includes a picture with an impressive very thick smoke "mushroom" clearly reaching into the stratosphere. The

pyro-Cb burst most likely responsible for our observation occurred at 51.0º N and 120.8º W, at times between 19 UTC on 30 June and after UTC midnight. The Seattle radar yields a maximum altitude of 17.3 km at 2:12 UTC. This altitude exceeds that of the largest peak in Fig. 7.

Unfortunately, we could not find a suitable "aerosol curtain" in the images derived from the measurements of CALIOP visualizing the fire plume right after the event. However, we inspected numerous graphics of the Multi-

angle Imaging SpectroRadiometer (MISR, https://misr.jpl.nasa.gov/) and the Micro-Pulse Lidar Network (https://mplnet.gsfc.nasa.gov) to establish the path of the smoke.

The interpretation was hardened by running numerous HYSPLIT 315-h forward and backward trajectories for different start altitudes with the GDAS option. As in the case of the 2017 fires there is no homogeneous flow pattern, the path of the trajectories strongly varying with altitude. In Fig. 9 we show ensemble trajectories

initiated at 15.6 km a.s.l. and 23 UTC (24 CET) above IMK-IFU, the position of the largest aerosol spike in Fig. 8 and start positions varied by one grid point. As mentioned there is a strong sensitivity on start time and position. One trajectory bundle leads backward towards British Columbia and three trajectories from this bundle end not far from the position of the main pyro-Cb 261 h backward in time (1 July, 2 UTC, almost within the pyroCb period mentioned above). The trajectory results for the aerosol peaks between 12.3 km and 13.8 km are

less perfect.





The trajectory results provide evidence that the main burst of the plume directly passed over our site during the first round around the globe. This explains the very sharp structure of the spikes.

**3.3 Volcanic eruptions in June 2019 and January 2022**

*Eruptions in 2019*

In June 2019 there is the interesting case of an almost co-incident volcanic eruption in the tropics and in the mid-latitudes. According to information from the Global Volcanism Program (GVP) of the Smithsonian Institution (https://volcano.si.edu/index.cfm) the Raikoke volcano on the Kuril Islands (48.292º N and 153.23º E, summit 558 m a.s.l.) erupted on 21 June 2019. Cameron et al. (2020) reported strong $SO_2$ after Raikoke eruption at 24 km, tapering off within four months. Boone et al. (2002) and Knepp et al. (2022) discuss the satellite

observations of $SO_2$ and sulfate, the latter having been observable until the following spring, obviously at a different cut-off level.

Subsequently, Ulawun (New Guinea, 5.05º S and 151.33º E, summit 2334 m a.s.l.) violently erupted on 26 June 2019. The maximum altitude reached was 19.2 km which should be within the tropical stratosphere. A plume component at 16.8 km is reported to having drifted north-east- to north-westward. Both eruptions were studied

by Kloss et al. (2021).

Our first observation of two small aerosol peaks just above the tropopause that could be related to the Raikoke plume took place on 19 July 2019. The signal could not be inverted because tropospheric clouds strongly attenuated the signal. From 23 July to September first a number of spikes appeared between the tropopause and 19.5 km, i.e., less than the maximum plume height reported by Cameron et al. (2020). The spiky distribution

gradually changed to a less structured hump (Fig. 10). The maximum backscatter coefficient above the respective tropopause was $4.18 \times 10^{-7}$ m$^{-1}$ sr$^{-1}$ on 23 July (the unidentified spike at 12.8 km with $7.53 \times 10^{-7}$ m$^{-1}$ sr$^{-1}$ could be due to a cirrus). This is quite high for stratospheric aerosol. The maximum integrated backscatter coefficient was calculated for 22 August ($5.4 \times 10^{-4}$ m$^{-1}$ sr$^{-1}$ , 693.4 nm). It exceeded those for the eruptions of Mt. St. Helens and Alaid in the early 1980s and is the third highest in our series, following the maxima for

Pinatubo and El Chichon. The temporary minimum of the integrated backscatter coefficient in September is 2019 caused by very high tropopause levels up to 15 km. The structured contributions after the Raikoke eruption gradually tapered off until January 2020, reaching a maximum altitude of 22 km. They were followed by a smooth hump with elevated aerosol (up to $5 \times 10^{-8}$ m$^{-1}$ sr$^{-1}$) ending between 20 and 21 km.

As mentioned in Sect. 3.2 Ohneiser et al. (2021) emphasize the additional influence of Siberian fires during that

summer. We defer to the view in that paper, derived from differences in the retrieved extinction-to-backscatter ratios. Ohneiser et al. (2021) conclude that the volcanic portion of the aerosol is mostly that at higher altitudes.

It is interesting that the decay of the integrated stratospheric backscatter coefficient is slower than after most mid- and higher-latitude volcanic eruptions (Fig. 3). We speculate that this could be due to the larger area of these strong fires in comparison with a volcanic point source and longer-lasting burning, leading to wider

horizontal spread of the particles.

There is some indication that we are able to distinguish between contributions from both 2019 eruptions in our data. This distinction is based on a delayed arrival of what we could ascribe to the tropical component (see



(Jäger, 2005) for the eruptions of El Chichon and Pinatubo). On 20 December 2019 a sudden rise of the upper boundary of the stratospheric aerosol to clearly beyond 30 km started. Such a rise would require a Brewer-

Dobson-type lifting of the tropical air mass.

In Fig. 11 we give a few examples of smoothed scattering ratios from the times before, during and after that period of enhanced aerosol. Up to 35 km the uncertainty of these values stays within ±0.03. The noise level above 35 km becomes rather high (up to ±0.1) because the scattering ratio implies a division by the strongly decreasing molecular backscatter coefficient. It is, therefore, difficult to determine precisely the cut-off altitude if

it lies between 35 and 40 km. The scattering ratio was almost constant up to the upper boundary and typically ranged between 1.04 and 1.10. This indicates a rather homogeneous aerosol distribution in agreement with the idea of long transport times.

Figure 12 shows the time series of the principal aerosol upper boundaries determined from the retrieved profiles of the backscatter coefficients. On 26 April 2021 the layer extension to beyond 30 km completely disappeared

(Fig. 13). Apart from occasional very small peaks this remained unchanged until the end of the measurements included in this paper.

These observations are further discussed in Sect. 4, in comparison with the Pinatubo results (Jäger, 2005).

***Hunga Tonga 2022***

The most violent volcanic eruption in recent history occurred on 15 January 2022, lasting just 11 h. The Hunga

Tonga Hunga Ha'apai submarine volcano (20.55º S, 175.4º W) injected material, including huge amounts of steam (Schoeberl et al., 2022, Xu et al., 2022; Vömel et al., 2022), into the stratosphere up to as high as 58 km, far beyond the 40 km reached by the Pinatubo eruption (Proud et al., 2022; Taha et al., 2022). The bulk of the plume circulated the globe in the southern hemisphere at altitudes between 20 and 30 km. Most of the poleward expansion occurred in the southern hemisphere. However, some material also reached the Arctic. In the tropics

maxima between 20 and 45 ppm of water vapor were detected between 25 and 26 km by sonde ascents (Vömel et al., 2022).

But also at high latitudes observations of the plume were made. Taha (2022) traced an aerosol layer observed at 83º N, 29º E and 21 km on 4 April 2022 back to the Hunga Tonga cloud. Khaykin et al. (2023) verified northward transport to 80º N within three to four months by using satellite and lidar measurements. The altitude

range was 20 to 25 km (see also (Mishra et al., 2022)).

The only of our measurements showing a conspicuous feature in spring and summer 2022 was made on 29 June (Fig. 14). A small aerosol peak, not seen in other measurements during that period occurred at 22.75 km. This is between the altitudes in the observations of the Tonga plume at Haute Provence (Southern France) and Kühlungsborn (North Germany) as presented by Khaykin et al. (2023).

In order to identify the advection path for the peak in Fig. 14 we calculated HYSPLIT 315-h backward trajectories for re-analysis data and in ensemble mode (Fig. 15). The air mass arrived from the east and passed over China on 17 June. Figure 16 shows aerosol curtains of the Ozone Mapping and Profiler Suite (OMPS) limb sounder for the start and end times of the trajectories. On 29 June the orbits closest to Garmisch-Partenkirchen (UFS) a feature with elevated aerosol extinction ratio around 22.5 km conforms to the lidar observation. This is





just indicative of the presence of Tonga particles. However, the Asian orbit on 17 June (lower panel) reveals a direct connection to the Tonga plume depicted in dark colour at the same altitude across the equator.

No peak around this altitude was seen again before the measurement on October 5. Starting in October elevated aerosol was found around and below 20 km and below 25 km. In Fig. 17 we show the scattering ratios for four selected measurements from the period between October 2022 and February 2023. The relative importance

varies with time, a minimum was found for the end of December and January. This (and the changing tropopause) explains the strong variation of the integrated backscatter coefficients in Fig. 3.

On 19 October for the first time a particularly pronounced peak structure was retrieved. This is confirmed by an OMPS aerosol curtain (Fig. 18). We prefer to display the extinction coefficients instead of the extinction ratio for more clearness. This reduces the sensitivity for the aerosol structures in the northern hemisphere. As in Fig. 16

the stratospheric aerosol maximizes in the tropics and the southern hemisphere, as one would expect from the position of the Tonga archipelago. The x-shaped crosses at 25 km indicate a separate aerosol layer, slightly above the lidar peak.

Also during the following months the lidar aerosol maxima are located below the OMPS crosses. Not always stars exist, in agreement with lower structures in the UFS backscatter coefficients. Figure 19 shows the situation

for 12 February 2023: The OMPS aerosol below 20 km had grown considerably, which is confirmed by the pronounced peak for 12 February in Fig. 19. The corresponding OMPS extinction ratio (not shown) exceeds the colour scale in that image.

Motivated by the results of Vömel et al. (2022) we inspected the relative humidity (RH) distribution in the Munich radiosonde data. Normally, the RH values of the RS41 sonde launched by DWD decrease to 1 % within

a few kilometres above the tropopause, 1 % mostly being the lowest value listed. Starting in October 2022 RH ≥ 2 % became more and more frequent. By February 2023, the maximum RH ranged between 3 % and 5 %, 12 % on 13 February. The range of particularly elevated humidity was located clearly above the aerosol maximum.

The elevated RH values are strongly indicative of the Tonga plume. However, the RH maxima are located above the aerosol maxima. However, Khaykin et al. (2022) demonstrated that the humidity layers may differ in altitude

from layers with depolarized particles.

Measurements with our Raman lidar (Klanner et al., 2021) in February 2023 indicate an increase of the water-vapour mixing ratio above 17 to 20 km, with an indication of further rise towards higher altitudes. However, the laser power was low which resulted in strongly enhanced uncertainty starting in this altitude range. and we prefer not to emphasize these findings.

**4. Discussion and Conclusions**

With the new lidar at UFS the long-term Garmisch-Partenkirchen stratospheric aerosol series has been continued since March 2016. The signal-to-noise ratio of the system has greatly improved allowing a better performance for 7.5-m vertical bins than previously with 75-m bins. The data evaluation, based on a Klett algorithm, now starts at r = 45 km (h = 47.7 km), but this could be extended even to larger distances. The limit is given by the

NCEP pressure and temperature data used for the calibration of the aerosol backscatter coefficients that end before 55 km a.s.l.



The integrated aerosol backscatter coefficients (Fig. 3) are dominated by the contributions from the first kilometres above the tropopause. Here, particles from moderate mid- and high-latitude volcanic eruptions, pyroCbs, desert dust (Trickl et al., 2013) and aircraft emissions cause a pronounced variability, sometimes featuring a spiky structure. These aerosols are removed at short to moderate time scales by stratosphere-to-tropopause transport (e.g., tropopause folding, Fig. 6; Stohl et al., 2003) and dilution. Above 25 km the aerosol contributions in our data mostly disappear within less than 1.5 years. After the removal at low and high altitudes the maximum scattering ratio is typically observed around 20 km.

Vernier et al. (2013) concluded from satellite-based measurements that a calibration of an aerosol lidar with stratospheric capability must take place beyond 40 km. This is definitely true for tropical stations where the aerosol is likely to extend to higher altitudes than in the mid-latitudes. Indeed, the Mauna Loa lidar observations quite often show aerosol at 37 to 38 km (John Barnes, personal communication, 2021). However, for our mid-latitude station we normally find upper boundaries of the aerosol between 25 and 30 km. In any case, given the performance of the new system, we are now prepared for periods with minor amounts of aerosol reaching to at least 35 km as found during a one-year period in recent years (see below).

The background phase 1999 to 2008 was rather special. The particularly low integrated backscatter coefficients during that period yielded integrated backscatter coefficients down to about 40 % of the 1979 average background (horizontal line in Fig. 3) that were never reached again later on. Most likely, the 1979 background did not represent a minimum during that period because of the tropical Fuego eruption in 1974. The very remarkable aerosol depletion on some measurement days calls for more elaborate analysis of the reasons, such as troposphere-to-stratosphere transport (TST).

Indeed, TST was observed by us in a few cases in recent years and resulted in low aerosol up to a few kilometres above the tropopause (not presented here). For example, the occurrence of TST has been associated with upward transport in warm conveyor belts (WCBs; e.g., Stohl and Trickl, 1999; Trickl et al., 2003). Stohl (2001) and Madonna et al. (2014) estimated that overshoots of WCB air into the stratosphere can reach 10 % or more. In addition, frequent vertical exchange between the troposphere and the stratosphere occurs along the subtropical jet stream (Sprenger et al., 2003; Trickl et al., 2011).

Aerosol sources were not completely absent during the low-background period, in particular strong eruptions in the tropics (Massie, 2016), but obviously did not significantly influence our observations. Most relevant for our station are mid-latitude eruptions to at least 10 km (Table 1 of Trickl et al., 2013). However, mid-latitude events with layer tops of 12 km and more did not occur before 2006.

After the background phase there were two periods with clearly enhanced stratospheric loading, 2008 to 2012 and since 2017, which is the most spectacular phase since the Pinatubo eruption. In 2020 and early 2021 some aerosol extended to more than 35 km, tentatively ascribed to the tropical Ulawun eruption. This would be supported by Stenchikov et al. (2021) who report a maximum altitude of 35 km on the basis of model calculations and SAGE data (cited by these authors as: Thomasson and Peter, 2006), after a rise from initial 17-26 km (Winker and Osborn, 1992; Guo et al., 2004).

It is interesting to compare this case with the more violent burst of Pinatubo. We, thus, inspected the evaluated profiles for 1991 to 1995 and mostly found rather sharp cut-offs near 30 km. Resolvable aerosol backscatter



coefficients up to $2\times10^{-9}$ m$^{-1}$ sr$^{-1}$ (given the 75-m bin size chosen for the photon counter) rarely extend to more than 32 km.

The absence of discernible aerosol contributions beyond 32 km during the Pinatubo period suggests to be careful in the 2020 case. One possible explanation could be a temporary offset of the NCEP data. However, it is difficult to assume such a bias for more than a year.

As an additional stratospheric contribution to the 2019 eruptions Ohneiser et al. (2021) report large Siberian fires in July and August 2019. They observed the plume in the polar vortex up to 18 km by lidar measurements onboard the research vessel Polarstern between October 2019 and May 2020. The origin of the particles in these fires was concluded by a high lidar ratio of 85 sr at 532 nm. Without the planned high-spectral-resolution channel we could not fully answer the question on how much of the Siberian smoke passed over our station at

just 45.455º N.

The observations of enhanced stratospheric aerosol in the Artic could provide an answer to the question why the integrated backscatter coefficient decreased so slowly in 2020. Grooß and Müller (2021) report a particularly stable Arctic vortex and a pronounced ozone hole until early April 2020. This could have led to a retarded outflow of aerosol-loaded air from the vortex. Indeed, the HYSPLIT trajectories for our two measurement days 7

April and 15 April show a transition from almost circular vortex to one with a more folded structure. The tropopause was rather high in March and April which reduces the integrated backscatter coefficients, but Fig. 3 reveals an upward step.

Since 2017 a sudden increase of violent pyro-Cbs injecting smoke into the stratosphere has contributed to our observations. Indeed, Peterson et al. (2021) report an increasingly large stratospheric influence of pyroCbs.

Khaykin et al. (2020) report on Australian fires up to 35 km. In our time series (Fig. 3) the first pronounced contribution of a pyroCb was the Québec fire in May and June 1991 (Fromm et al., 2010) just preceding the arrival of the Pinatubo plume and, therefore, initially not correctly identified (Carnuth et al., 2002). The recent rather sudden rise in strong loading of the stratosphere with particles from biomass burning up to even more than 20 km suggests further research. It is interesting to note in this context that the area burnt in the U.S. discussed

by Trickl et al. (2013) has no longer increased since 2005.

During the period 2017 to present, discussed in this paper, there have been several opportunities to study the depletion of stratospheric aerosol. The depletion is mainly due to stratosphere-to-troposphere transport from the tropopopause region, dilution (Fromm et al., 2008) or advection of clean air masses. Unfortunately, the aerosol injections into the stratosphere were too frequent to allow us to observe a depletion down to the lowest values in

the time series. As obvious from Fig. 3 the stratospheric aerosol loss in the case of mid-latitude eruption of Mt. St. Helens in 1980 occurred within a single year. Also after the first British Columbia pyro-Cb in 2017 the recovery of the stratosphere occurred within less than one year. The decay after the Raikoke eruption was much slower. It is reasonable to assume that additional aerosol contributions reached the stratosphere during that period, such as the tropical Ulawun eruption. What we realized in 2021 and 2022 is that aerosol depletion took

place first at high altitudes and then also just above the tropopause.

We are glad that more and more sources of stratospheric aerosol can be identified by following satellite measurement curtains or transport modelling or a combination of both. A special success was the identification of the Tonga plume in our profiles. The tools meanwhile available on the internet, in particular transport models





such as HYSPLIT or FLEXPART (https://www.flexpart.eu/), make possible an interpretation of the observations
in much more detail than a few decades ago. Still, it is a challenge to follow plumes that have been in the
stratosphere for more than the two weeks for which transport modelling in the free troposphere and stratosphere
is applicable (e.g., Trickl et al., 2011; 2015) such as in the case of transport from the tropics to the mid-latitudes.
However, the growing information on strong aerosol sources allows one to determine the origin of pronounced
features in the retrieved aerosol distributions.

**5 Appendix**

*Rayleigh scattering*

The calculation of the Rayleigh backscatter coefficients can be done with a relative uncertainty of about 0.5 % in
the visible spectral range for a careful approach, if the atmospheric density is known with sufficient reliability.
The details of Rayleigh scattering as applied in the IFU lidar algorithms are described in a review prepared in
2013 for the NDACC Lidar Working Group ("ISSI Team", Leblanc et al. 2016a, b; meetings held at the Inter-
national Space Science institute, Bern, Switzerland) that is available in a revised version on the internet (Trickl,
2023). Because of the importance for the NDACC quality assurance we describe here a few important facts.

The total particle-free atmospheric scattering cross section is (in slight modification of Goody, 1964)

$$\sigma_R = \frac{24\pi^3}{\lambda^4 N^2} \frac{(n^2-1)^2}{(n^2+2)^2} F_K \tag{1}$$

with the refractive index $n$ of air, the air density $N$ and the King correction factor $F_K$. that implies the influence of
Raman scattering. We traditionally (Kempfer et al., 1994) take the refractive index of air from a computer
program reproducing the algorithm of Owens (1967) that provides the refractivity of air with an accuracy of
about eight decimal places, including $CO_2$ and humidity. The calculations of $n - 1$ are based on the Lorentz-
Lorentz formalism, which, consequently, was adopted also in Eq. 1. This ensures that $\sigma_R$ is constant as a function
of the air density to within $7\times10^{-7}$.

Introducing the isotropic part $\alpha$ of the polarizability the leading term can also be written as (e.g., She, 2001)

$$\frac{8\pi}{3} \frac{\pi^2}{\lambda^4} \alpha^2 = \frac{24\pi^3}{\lambda^4 N^2} \frac{(n^2-1)^2}{(n^2+2)^2} , \tag{2}$$

neglecting wavelength differences.

Bates (1984) lists $F_K - 1$ for wavelengths from 200 nm to 1000 nm with an estimated relative uncertainty of 1 %
(about 0.5 % visible spectral region). A least-squares fit to the $F_K - 1$ of these values using the expression

$$F_K - 1 = \sum_{i=0}^{2} p_i \lambda^{-2i} , \tag{3}$$

$\lambda$ in nm, yields the fit parameters (in brackets: relative standard uncertainties):

$p_0 = 4.69541179\times10^{-2}$ ($3.49\times10^{-3}$), $p_1 = 3.25031532\times10^{+2}$ ($1.06\times10^{-1}$), $p_2 = 3.86228507\times10^{+7}$ ($3.63\times10^{-2}$).



The $F_K - 1$ data are approximated by Eq. 1 mostly within clearly less than 1 % between 200 nm and 1000 nm,
respectively, implying a negligible relative deviation for $F_K$. The 532.24-nm backward differential cross section
without Raman contribution is $5.86612 \times 10^{-32}\,\mathrm{m^{-2}}$, the King factor $F_K = 1.048583$.

In the backward direction the correction factor is

$$F_K(\pi) = 1 + 0.7(F_K - 1) \tag{4}$$

The factor 0.7 differs from the value 1.0 used in many lidar applications. In the classical theory for polarized
scattered radiation and unpolarized detection the differential backscatter cross section for the Q branch
component of the central (Cabannes) line is

$$\frac{d\sigma_Q}{d\Omega}(\pi) = 0.25\frac{\pi^2}{\lambda^4}\alpha^2\left[0.7(F_K - 1)\right] = 0.25\frac{\pi^2}{\lambda^4}\alpha^2\left[\frac{7}{45}\frac{\gamma^2}{\alpha^2}\right], \tag{5}$$

$\gamma$ being the anisotropic part of the polarizability. For the quantum solution just a small correction is needed for
the first factor: For example, for T = 280 K we derive 0.25545 (nitrogen) and 0.26015 (oxygen), the sum over all
three branches being 1.0000 (Trickl, 2023). The sum of S and O branch relative line strengths is just slightly
below 0.75.

The influence of the interference filter in the far-field channel was estimated by calculating the nitrogen Raman
spectrum from spectroscopic data (Placzek and Teller, 1933; Herzberg, 1950; Trickl et al., 1993; 1995). A
relative contribution of the S and O branches of just 3 % of the full sum of about 0.75 of the relative O- and S-
branch line strengths was determined for the 0.5-nm width of the spectral filter in the far-field channel of the
receiver. This fraction is taken for the data evaluation in the filtered channel, but the influence in the retrieval is
small in comparison with the overall uncertainty.

***Uncertainties***

Although uncertainties of the backscatter coefficients have been determined in the past (Jäger, 2005) it is
important to give, for the first time, a few more details, from the current point of view. The highest contribution
to the uncertainty budget is caused by the calibration of the Klett retrieval. The sensitivity of the backscatter
coefficients to the far-field calibration of is extreme because of the mostly very small values of the Rayleigh
backscatter coefficients above the aerosol layer and the noise of the backscatter signal. Any deviation from the
best Rayleigh fit is interpreted as aerosol. In the range of zero aerosol backscatter coefficients typically down to
30 km the result must be perfectly centred in the noise (Fig. 1) in order to avoid a bias at altitudes below 30 km
that can readily reach 10 % and more otherwise.

The uncertainty of the air density is rather low. During the period under consideration the RS92 and RS41
radiosondes from Vaisala have been used by the German Weather Service (DWD). Steinbrecht et al. (2008)
carefully examined the RS80 and RS92 sondes in twin flights. The RS92 sonde turned out to be more accurate
and we assume a similar performance for RS41. The relative uncertainties of RS92 for temperature and altitude
are clearly below 1 % and, thus, do not contribute significantly to that of the air density. The pressure uncertainty
matters most at low pressures. Between 100 and 3 mbar it is specified as 0.3 mbar which means a relative
uncertainty of 3 % for the air density at the beginning of the calibration range of the lidar.



However, Rayleigh backscatter profiles calculated from the corresponding midnight sonde and the noon NCEP data have perfectly matched the photon-counting backscatter profiles for all low-noise measurements. The photon-counting channel is virtually free of any artefact. The high reliability could be further hardened by the one-hour temperature measurements even exceeding the range of the NCEP data (Klanner et al., 2021). Up to 53 km a.s.l., where the NCEP data ended in that case, the agreement with the lidar-based temperature was within 2 ($\leq$ 35 km) to 4 K (53 km). In the case discussed the temperature deviation was caused by a positive altitude offset of the NCEP data growing from 0 to 2 km between 40 and 53 km. More recent measurements demonstrated similar to better agreement. Therefore, the relative uncertainty of the NCEP data is of the order of 1 % and less.

Another source of uncertainty is the variability of stratospheric ozone. The maximum monthly mean ozone density in the MOHp analysis is located at 22 km, where we also see largest influence in the retrievals shown in Fig. 2. The monthly standard deviations evaluated for the MOHp ozone-sonde data range between 4.2 % (October) and 8.6 % (February). This yields a contribution much smaller than the overall uncertainty of the lidar retrieval.

The influence of the lidar ratio on the result of a stratospheric retrieval is small and, as mentioned, is based on values close to the recommended ones. However, there is a strong difference in the case of the extreme biomass-burning case on 25 August 2017. For this day we used a lidar ratio of 70 sr$^{-1}$ in the thin layer (Sect. 3.2) as determined by Ansmann et al. (2018). The backscatter coefficients below the plume decreased by about 20 % and then matched those above the layer.

For the integrated aerosol backscatter coefficients the chosen position of the tropopause is crucial because the highest contributions occur in the tropopause region. Mostly, the Munich tropopause looks very reasonable. However, as pointed out above, refinement is sometimes necessary. Without additional aerosol features at higher altitudes the highest values of the backscatter coefficients are found in the tropopause region. Thus, the choice of the tropopause is made with care (Sect. 2.2), based on the observations. The uncertainty of the integrated backscatter coefficients due to that of the tropopause normally does not exceed 10 %.

Cirrus clouds no longer influence the far-field signal with the 7400 PMT. In the past, signal induced nonlinearities were observed in the stratosphere if the EMI PMTs were overloaded even by big cirrus spikes in the tropopause region. Multiple scattering effects must be taken into consideration (Reichardt and Reichardt, 2006). They, indeed, exist in the case of thick cirrus clouds in the 313-nm channel of our ozone DIAL, but could not be verified in the green channel of the system described here except for a few extreme cases.

Leblanc et al. (2016b) derived a very complex approach to the determination of uncertainties. We strongly reduce the complexity by parametrizing the uncertainty u as

$$u = \sqrt{u_0^2 + (u_1 \frac{r^2}{r_{ref}^2})^2 + (u_2 S(r))^2}$$

The three coefficients $u_0$, $u_1$ and $u_2$ are adjusted by sensitivity analyses, separately for all three data channels taken (see Fig. 1 and the related explanations in Sect. 2.2).



We rather conservatively assume a minimum relative uncertainty of 15 % of the aerosol backscatter coefficient
($u_2$) until more experience is available. This approach chosen has entered the uncertainties archived in the
NDACC data base for the measurements since 2012.

## 6 Data availability

The 532-nm backscatter coefficients retrieved from the lidar measurements have been archived in the NDACC
data base (actual web address: https://www-air.larc.nasa.gov/missions/ndacc/data.html#). Until January 2016, the
station name is "Garmisch", afterwards "Zugspitze". The backscatter coefficients from 2000 until January 2016
are also stored in the EARLINET data base.

## 7 Author statement

TT evaluated the data. He interpreted the observations and wrote the paper, assisted by HF, MF and HJ. HV
carried out the lidar measurements since 2016. HJ provided the information on the Pinatubo period. HV and MP
built and optimized the new lidar at UFS, including the remote system control.

## 8 Competing interests

The authors declare that they have no conflict of interest.

## Acknowledgements

The authors thank Hans Peter Schmid for his support to continue the measurements. They acknowledge the
appreciation of this effort by their NDACC and EARLINET colleagues. It is important to mention the
contributions of Walther Carnuth († May 2021), Helmuth Giehl and Stefan Biggel in different stages of the
measurements. The technical improvements would not have been possible without the contributions of Werner
Funk, Bernd Mielke, Heinz Josef Romanski and Bernhard Stein. Wolfgang Steinbrecht provided ozone data
from the soundings at Hohenpeißenberg and was available for numerous discussions. Ludwig Ries made
available the UFS ozone data. This work has contributed to NDACC and EARLINET, the latter now being a part
of the European research infrastructure ACTRIS.
The publication of this article has been supported by the Helmholtz Association within its open-access initiative.



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



**Table 1.** Some operating conditions of the new lidar system

|  | *Period* | *number of laser shots* | *detection mode* |
|---|---|---|---|
|  | 2016 – Sept. 2017 | 10000 | analogue |
| 1100 | Oct. 2017 – 19 Feb. 2018 | 20000 | analogue |
|  | 25 Feb. 2018 – Apr. 2018 | 50000 | analogue; April: photon counting |
|  | May and June 2018 | 20000 | analogue and (June) photon counting |
|  | July 2018 – Aug. 2019 | 40000 | analogue and photon counting |
|  | Sept. 2019 | 100000 | analogue (failure of the counting system) |
| 1105 | Oct. 2019 – June 2020 | 40000 | analogue and photon counting |
|  | Since July 2020 | 100000 | analogue and photon counting |





**Figures:**

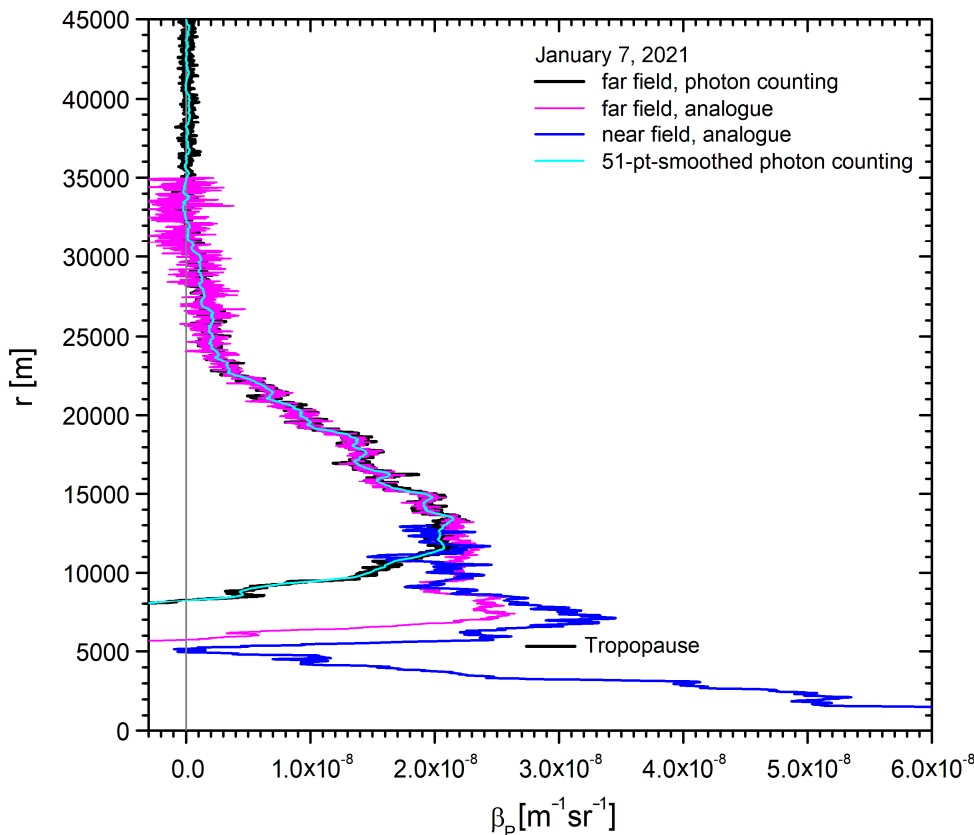

**Fig. 1.** Backscatter coefficients from the Klett inversions of the data of the near- and far field detection channels in the evening of 7 January 2021 (100000 laser shots); the values are displayed with bin sizes of 7.5 m (old system: 75 m, same noise amplitude up to 40 km). The photon-counting values are smoothed with a ±25-bin sliding average that reveals the high far-field performance of the system. Please, note the low wintertime Munich tropopause at r = 5.3 km (h = 8.0 km). r = 0 m corresponds to 2675 m a.s.l. (laboratory at UFS). The aerosol

below 5 km was advected from Ukraine and Turkey.



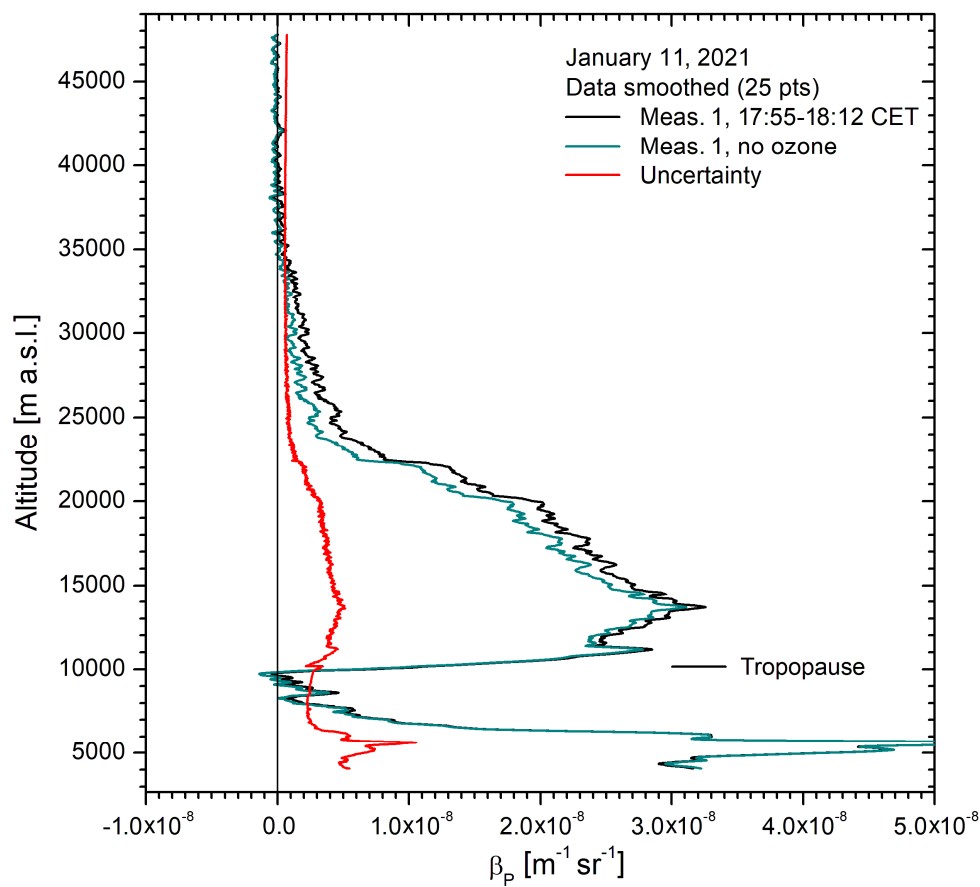

**Fig. 2.** Comparison of the retrievals with (black curve) and without (blue curve) ozone correction (11 January, 2021); the data were smoothed with a running arithmetic average over ±12 bins (±90 m).


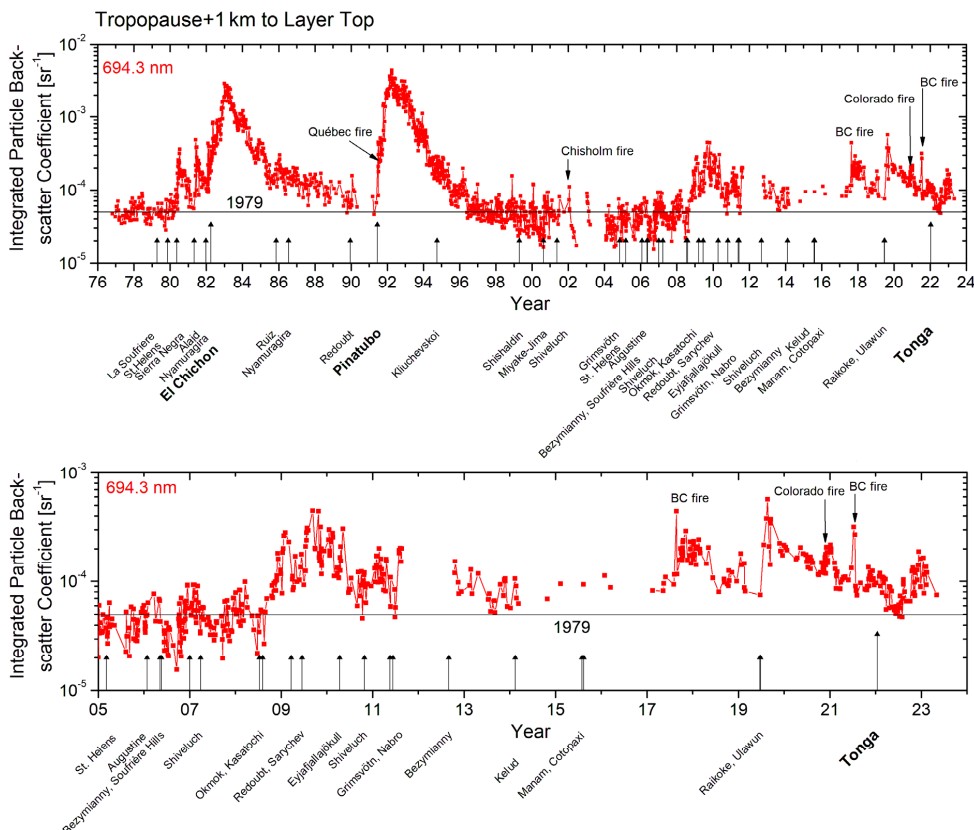

**Fig. 3.** Upper panel: Time series of the integrated stratospheric backscatter coefficient from the lidar measurements at Garmisch-Partenkirchen: The backscatter coefficients are integrated from 1 km above the tropopause to the upper end of the layer. Several minor events are marked in addition: two PSC observations, the Québec fire (Q) and the Chisholm fire (Ch).

Lower panel: Section of upper panel from 2005 to 2021



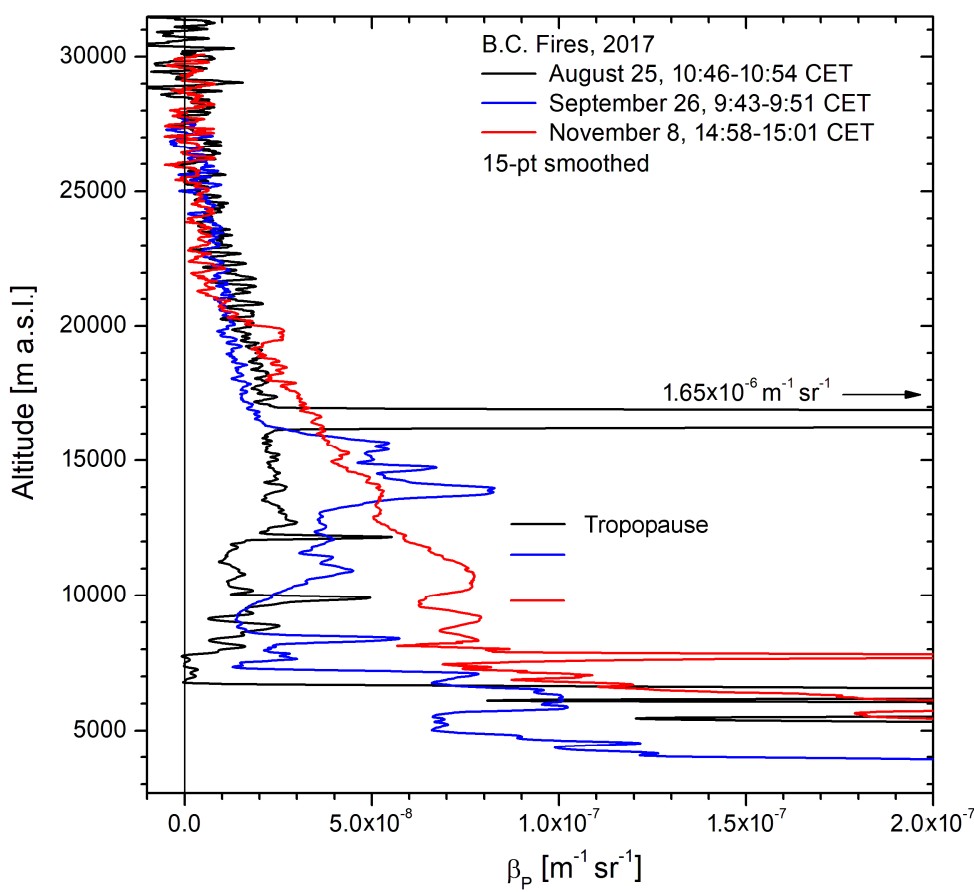

**Fig. 4.** Three examples of 532-nm backscatter coefficients following the British Columbia (B.C.) fires in 2017; the data are slightly smoothed (sliding arithmetic average over ±7 bins) because of elevated noise due to analogue data acquisition, daytime conditions and just 20000 laser shots. The corrected Munich tropopause altitudes of the Munich are marked in the colours of the corresponding backscatter coefficients.



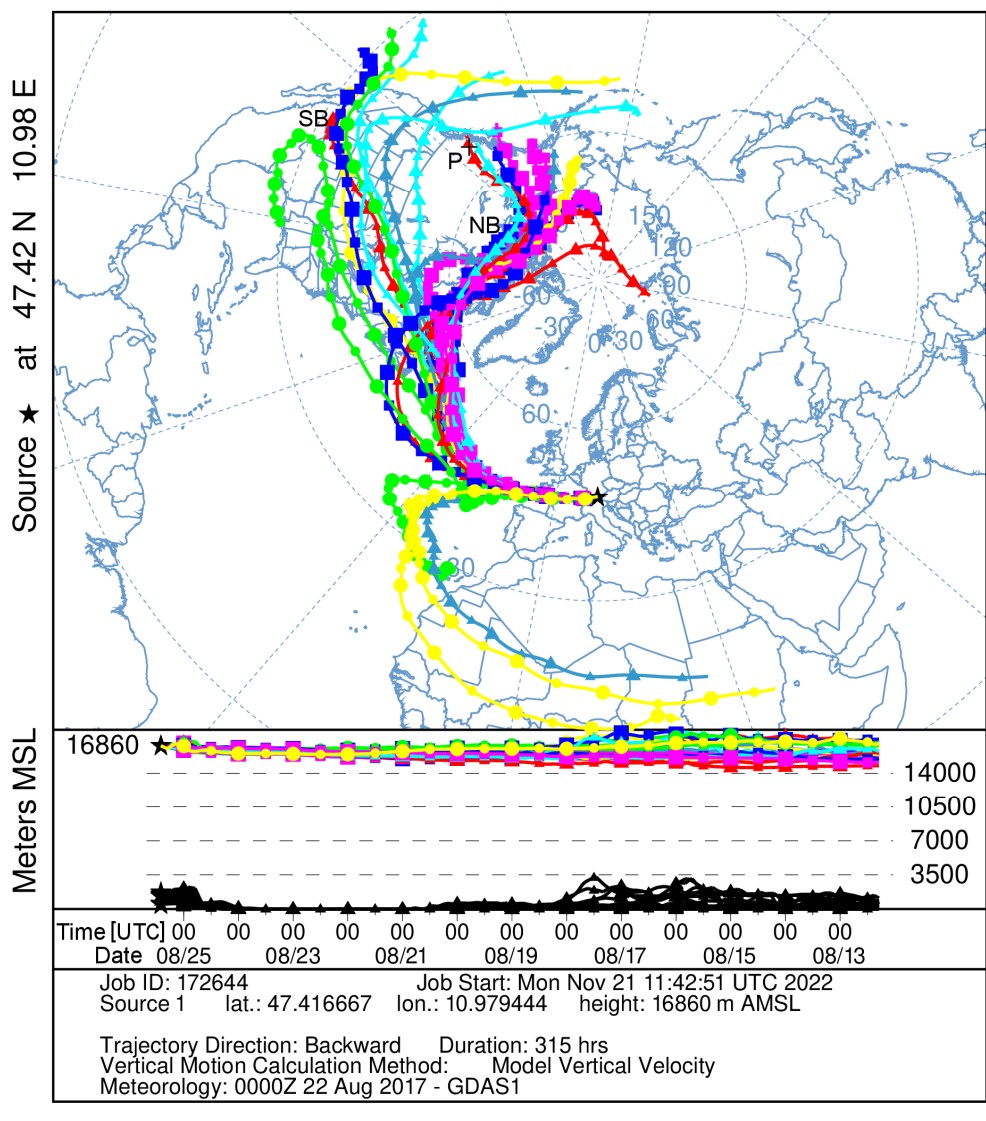

**Fig. 5.** HYSPLIT 315-h backward ensemble trajectories initiated at 16800 m a.s.l. above Garmisch-Partenkirchen; black asterisk (labelled with P) marks the approximate area of the pyroCbs.


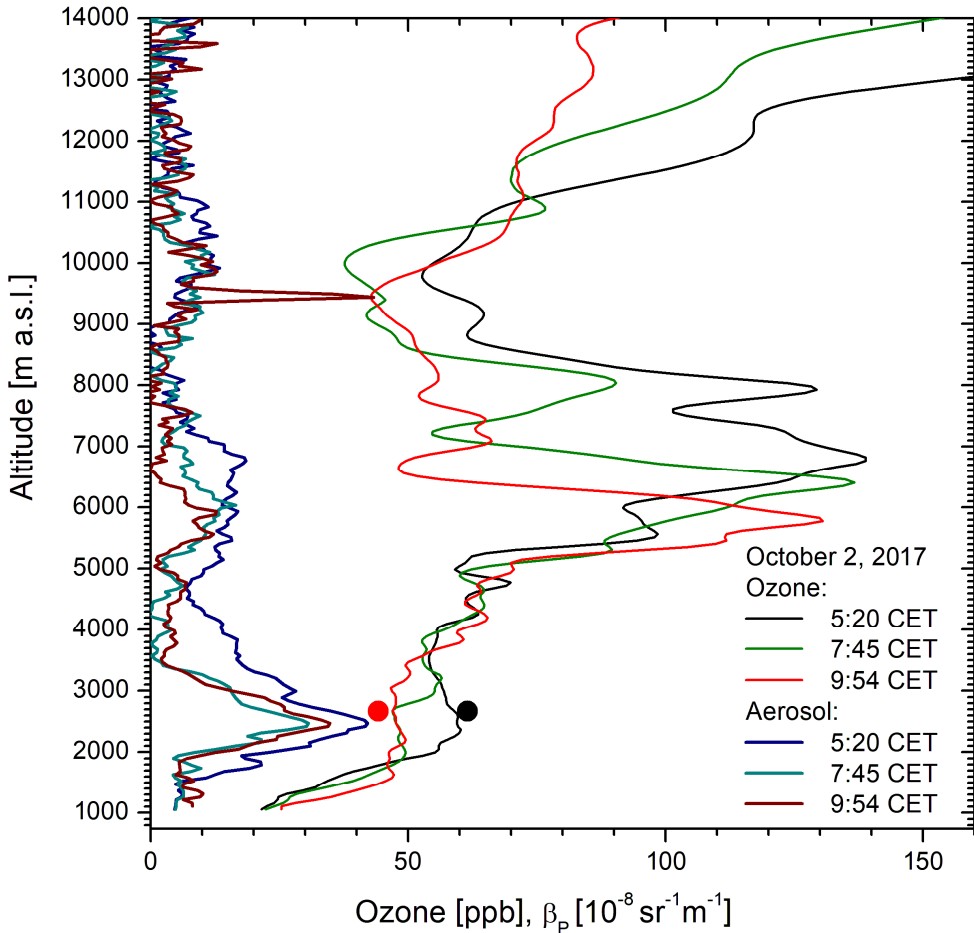

**Fig. 6**. Ozone mixing ratios and 313-nm aerosol backscatter coefficients derived from measurements of the ozone DIAL at IMK-IFU on 2 October 2017, showing aerosol in a descending intrusion layer giving rise to a 313-nm aerosol backscatter coefficient of almost $2\times10^{-7}$ m$^{-1}$ sr$^{-1}$ between roughly 5 and 7 km. The aerosol spike at 9.45 km could be a weak cirrus and mark the upper end of the troposphere. For comparison we also give the ozone mixing ratios measured at UFS at 5:30 and 10:00 CET (filled circles).



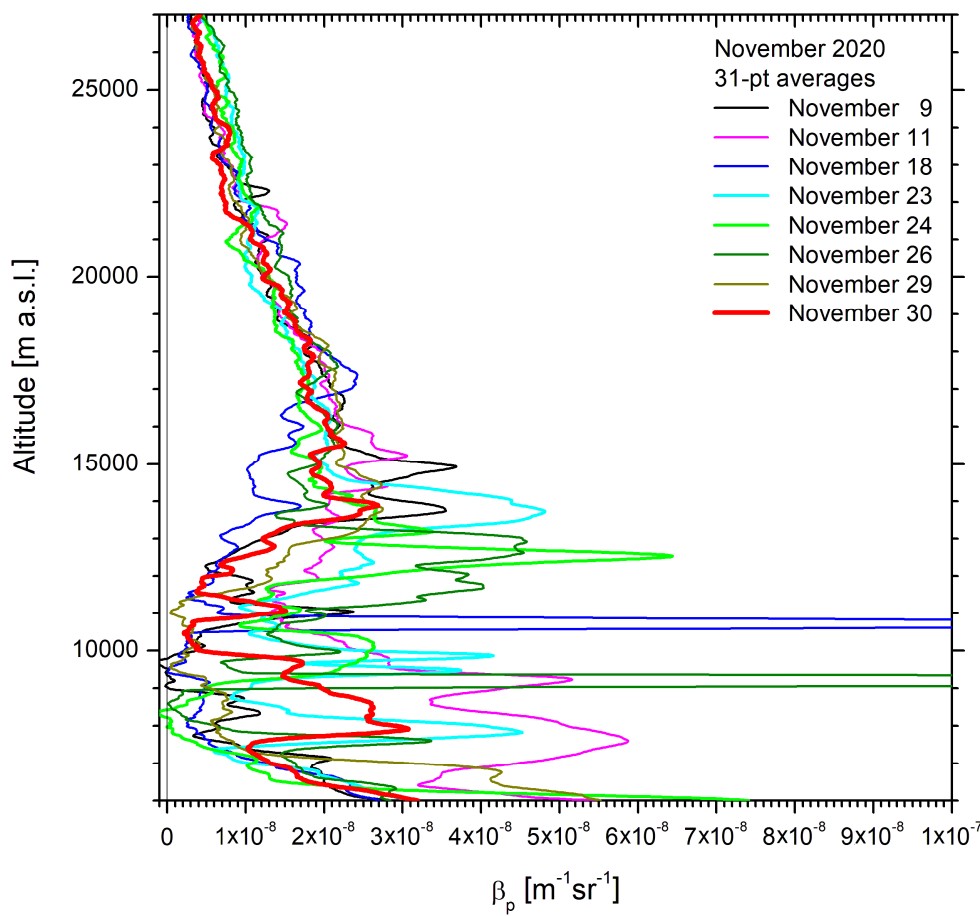

**Fig. 7**. Aerosol backscatter coefficients from the night-time measurements in November 2020 showing the influence of the fires in Colorado; the two spikes leaving the scale are caused by cirrus clouds. The corrected tropopause altitudes of the Munich radiosonde are 12.65 km, 13.13 km, 12.57 km, 11.41 km, 11.40 km, 10.93 km, 12.34 km and 13.06 km, respectively. See Fig. 1 for the situation after these plumes tapered off.




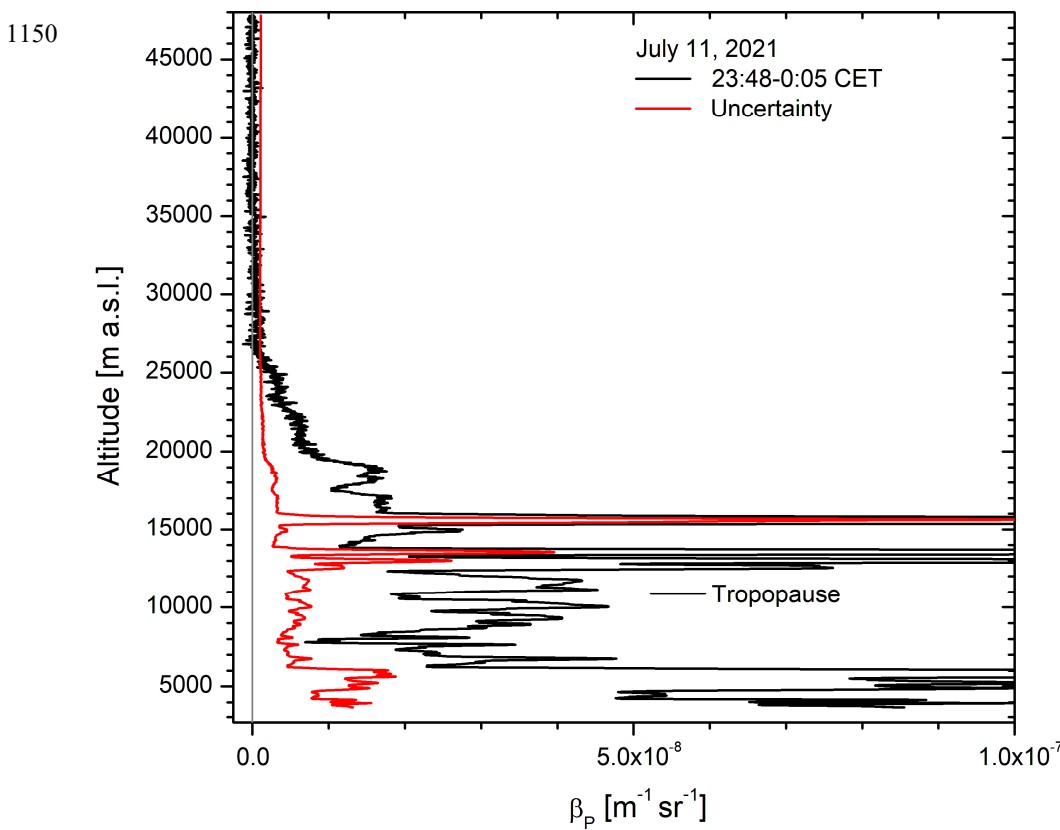

**Fig. 8.** 532-nm aerosol backscatter coefficients on 11 July 2021; the two maximum values are $2.63 \times 10^{-7}$ m$^{-1}$ sr$^{-1}$ at 13.5 km and $7.07 \times 10^{-7}$ at 15.6 km.



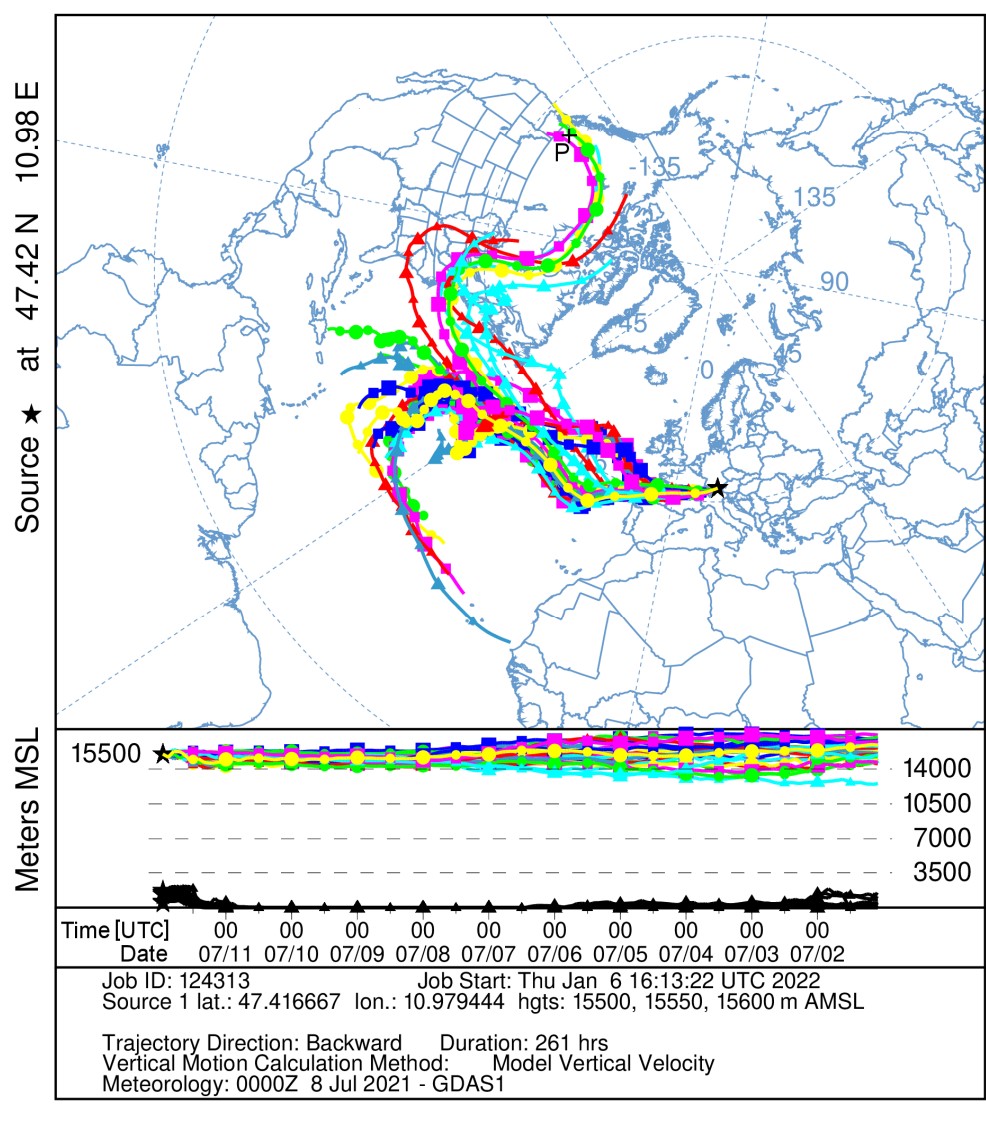

**Fig. 9.** HYSPLIT ensemble backward trajectories initiated at 15.55 km ± 0.05 km a.s.l. above UFS (Garmisch-
Partenkirchen) on 11 July 2021 (23 UTC); the duration of the trajectories is 261 h (see text); the most likely
pyro-Cb position at 51.0º N and 120.8º W is marked with a black cross (labelled with P).




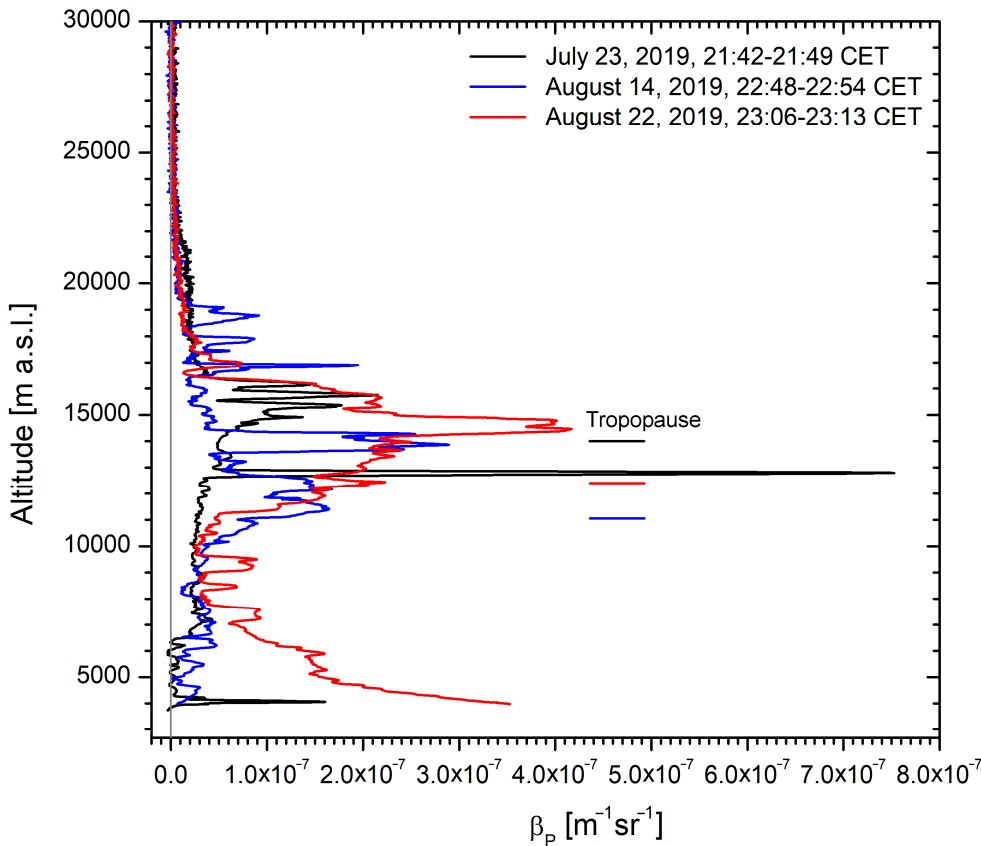

**Fig. 10.** High 532-nm aerosol backscatter coefficients following the Raikoke volcanic eruption in June 2019; the
tropopause levels are 13.89 km, 11.06 km and 12.40 km, respectively. We tend to assume that the spike at 12.8
km on 23 July was caused by a cirrus cloud because it is located below the tropopause. However, the Munich
relative humidity at that altitude was less than 30 %. The tropopause altitudes are marked in the colours of the
respective backscatter coefficients.



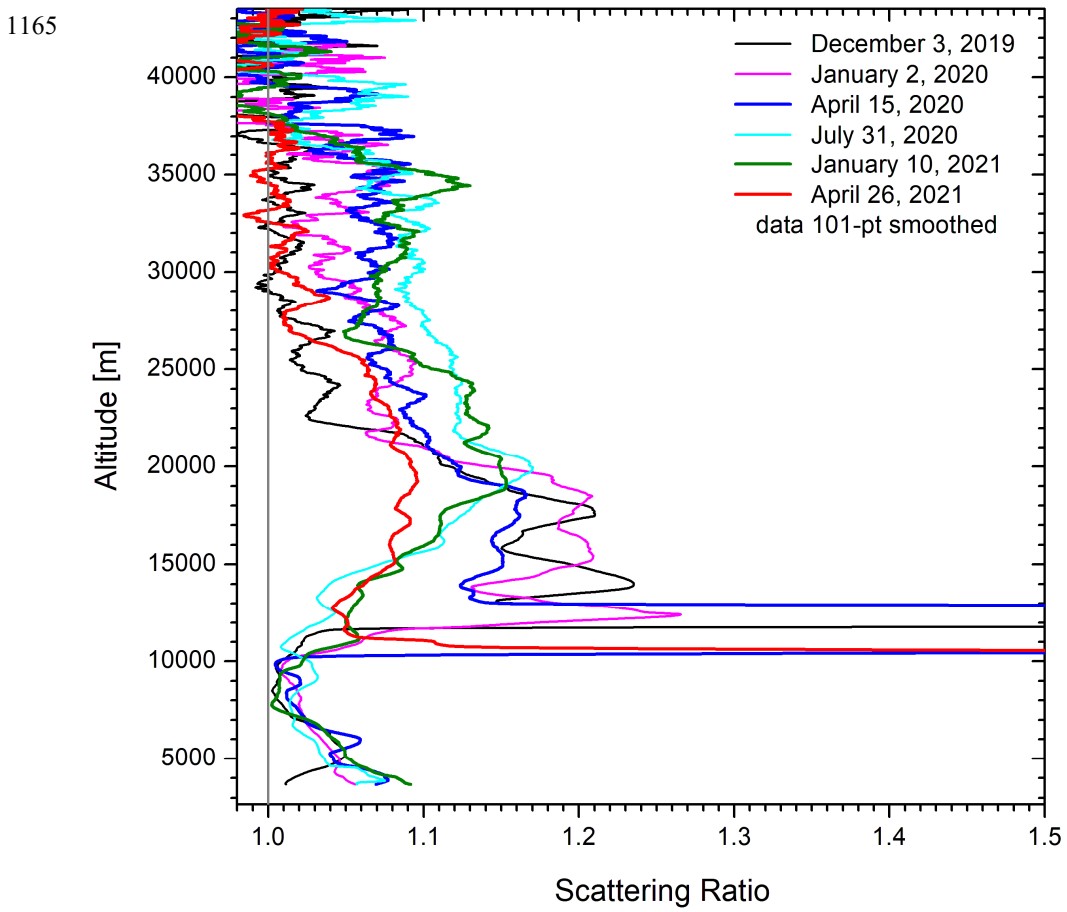

**Fig. 11.** 532-nm scattering ratios smoothed by gliding ±50-bin averages for selected measurements in 2019, 2020 and 2021; starting in January 2021 the aerosol layer expanded to more than 28 km a.s.l., possibly caused by northward propagation of the plume from the tropical eruption of Ulawun in the Brewer-Dobson circulation. The uncertainty of the values strongly grows above 35 km because the relative noise in the data starts to exceed the size of the aerosol features.




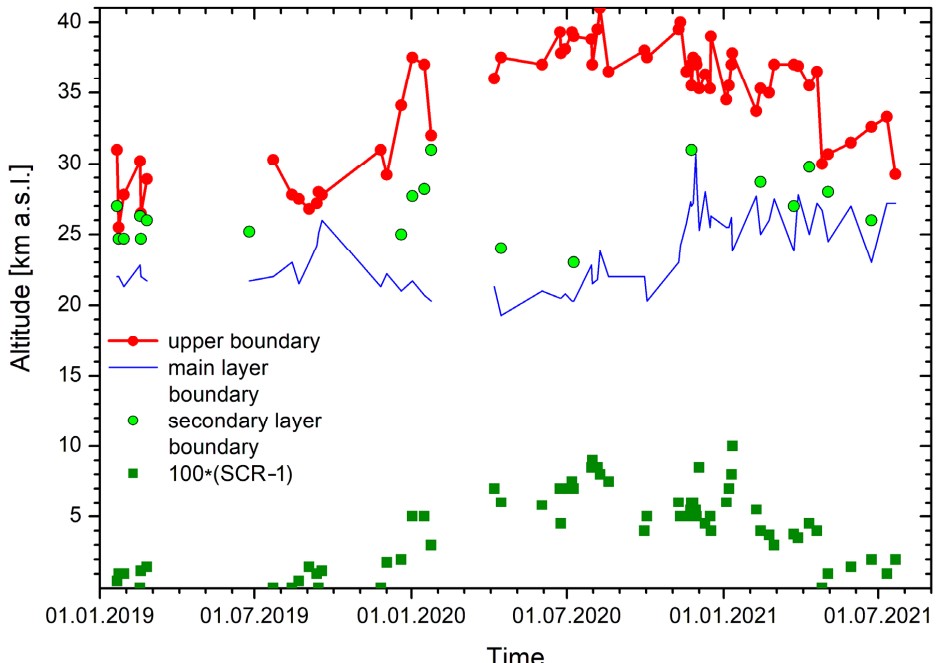

**Fig. 12.** Upper boundaries of the top aerosol layers after the two big volcanic eruptions in July 2019: "main
layer" (blue) means more a pronounced aerosol feature already present before that period. We speculate that the
rise of upper of the top boundary (red) was caused by northward propagation of the tropical eruption of Ulawun
in the Brewer-Dobson circulation. The average scattering ratio (SCR) above 30 km is slightly elevated.




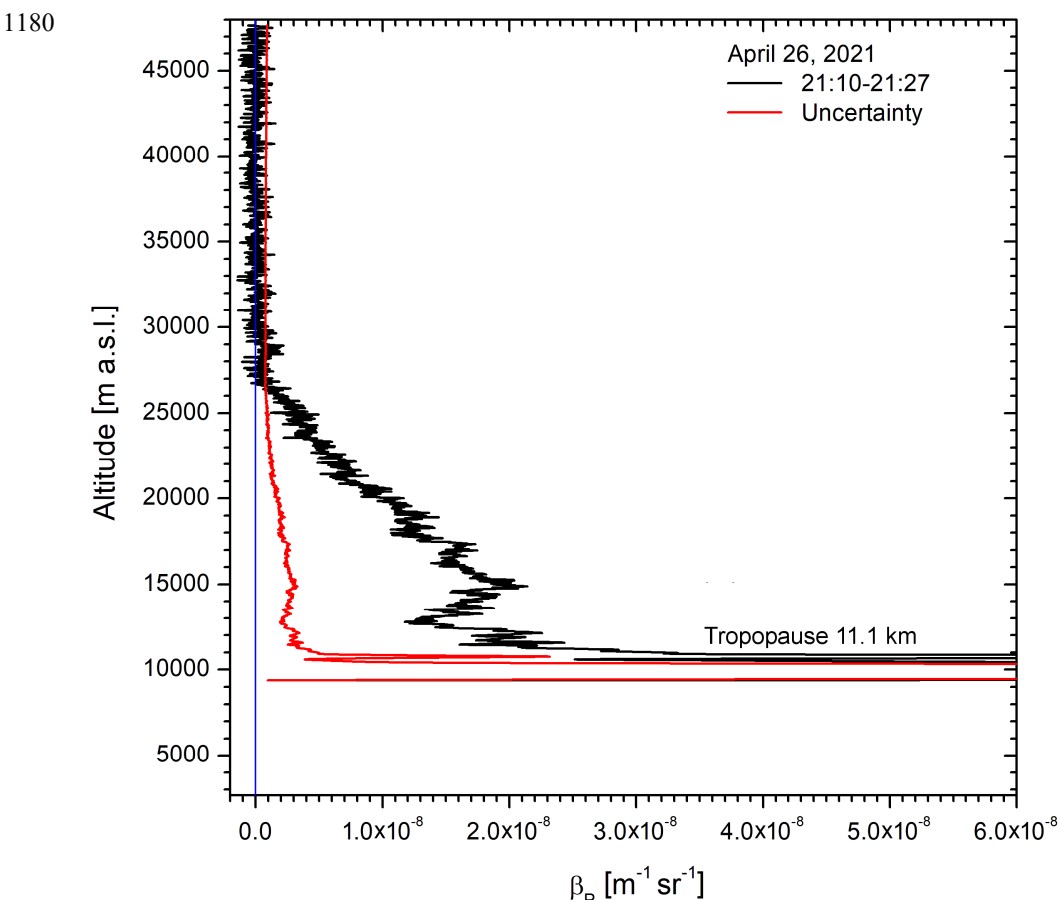

**Fig. 13.** The 532-nm aerosol backscatter coefficients for 26 April 2021.: There are no longer aerosol contributions above 30 km. The strong cirrus signal did not allow a reasonable data evaluation within the troposphere.




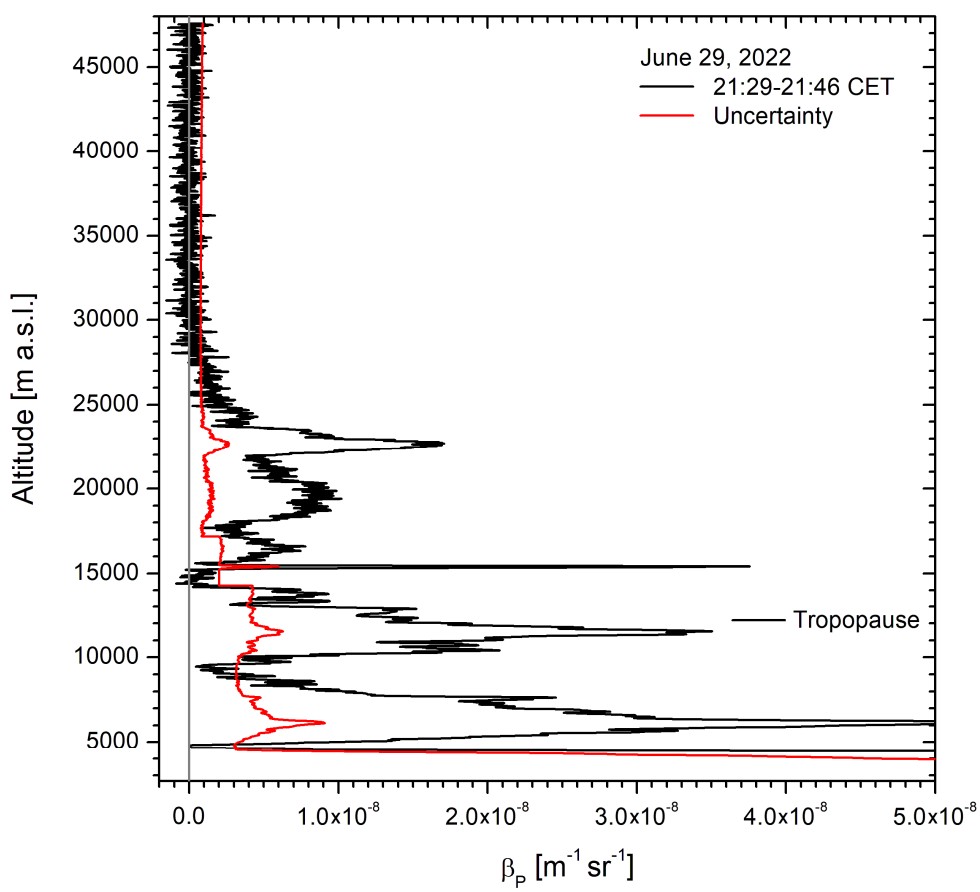

**Fig. 14.** 532-nm backscatter coefficients for the night-time measurement on 29 June 2022: The peak at 22.75 km is attributed to aerosol from the Hunga Tonga eruption on 16 January 2022 (see text). Please, note the rather low backscatter coefficients between 15 and 20 km that indicates the progress of aerosol removal from the stratosphere above our region after the recent events (see Fig. 12).




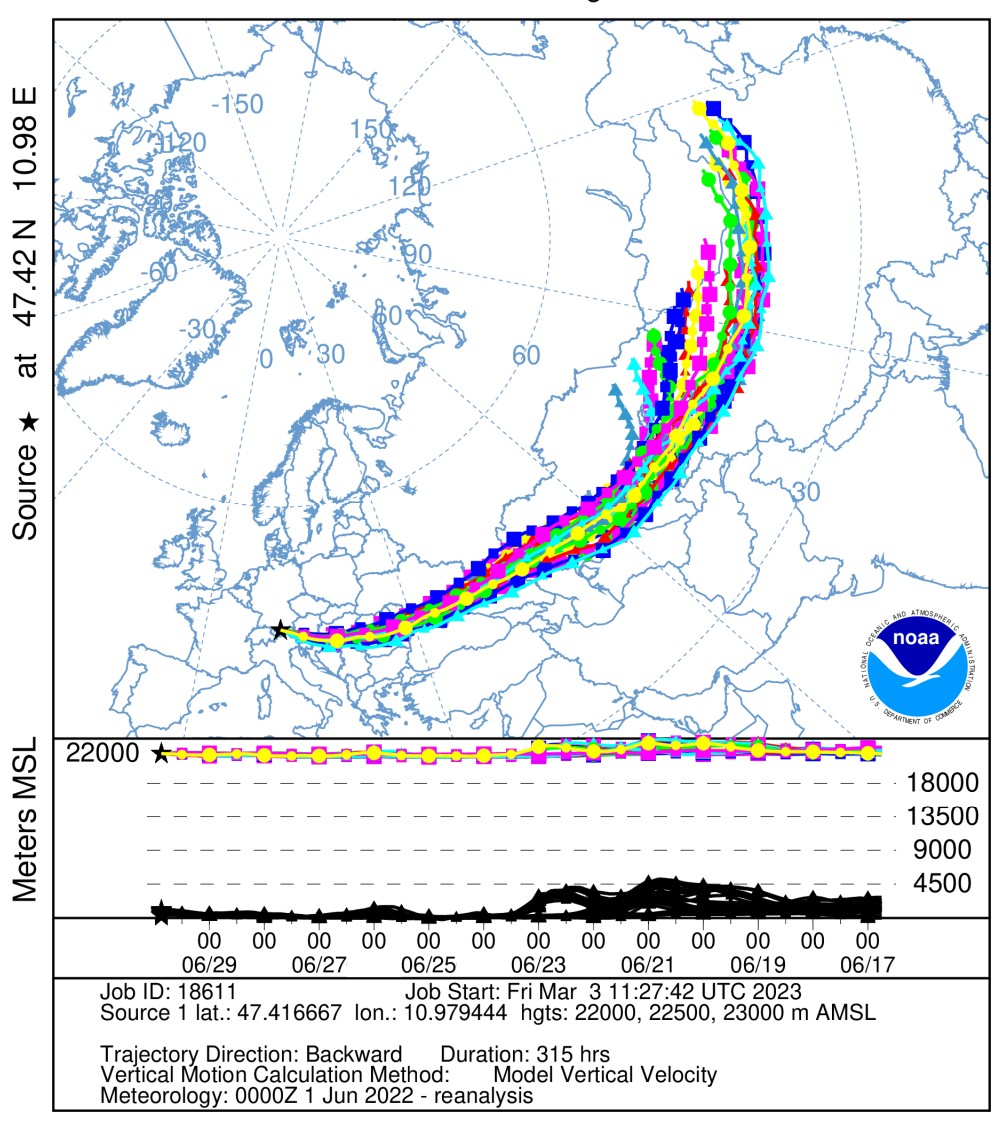

**Fig. 15.** HYSPLIT 315- h ensemble backward trajectories initiated above UFS on 29 June 2022 at 22:00 CET






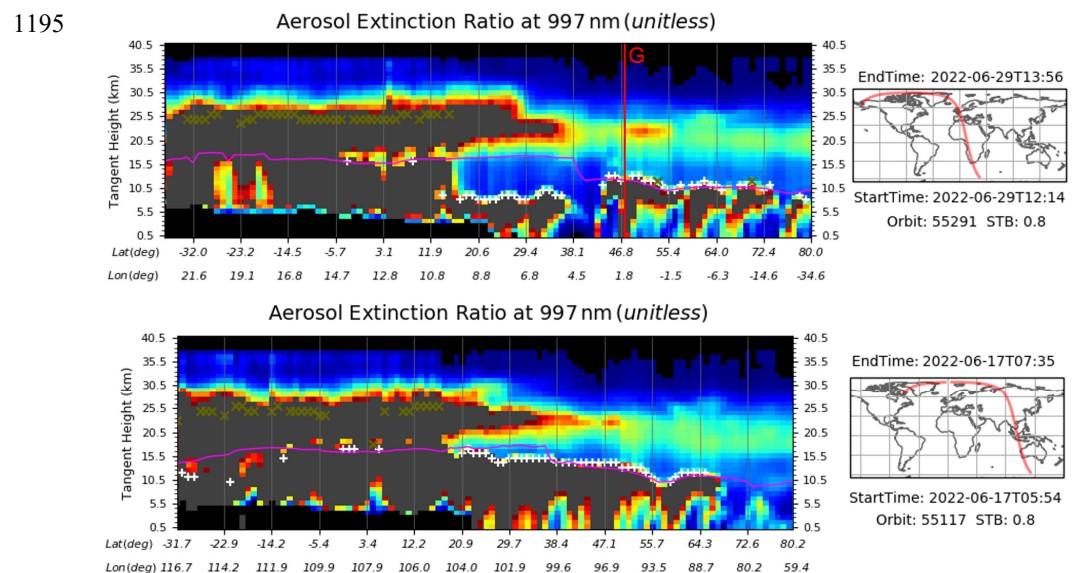

**Fig. 16.** OMPS vertical distribution of the 997-nm aerosol extinction ratio for orbits closest to UFS on 29 June and over East Asia on 17 June as indicated in Fig. 15; the vertical read line labelled by G marks the latitude of Garmisch-Partenkirchen. The panels to the right show the corresponding orbits of the satellite (red lines).




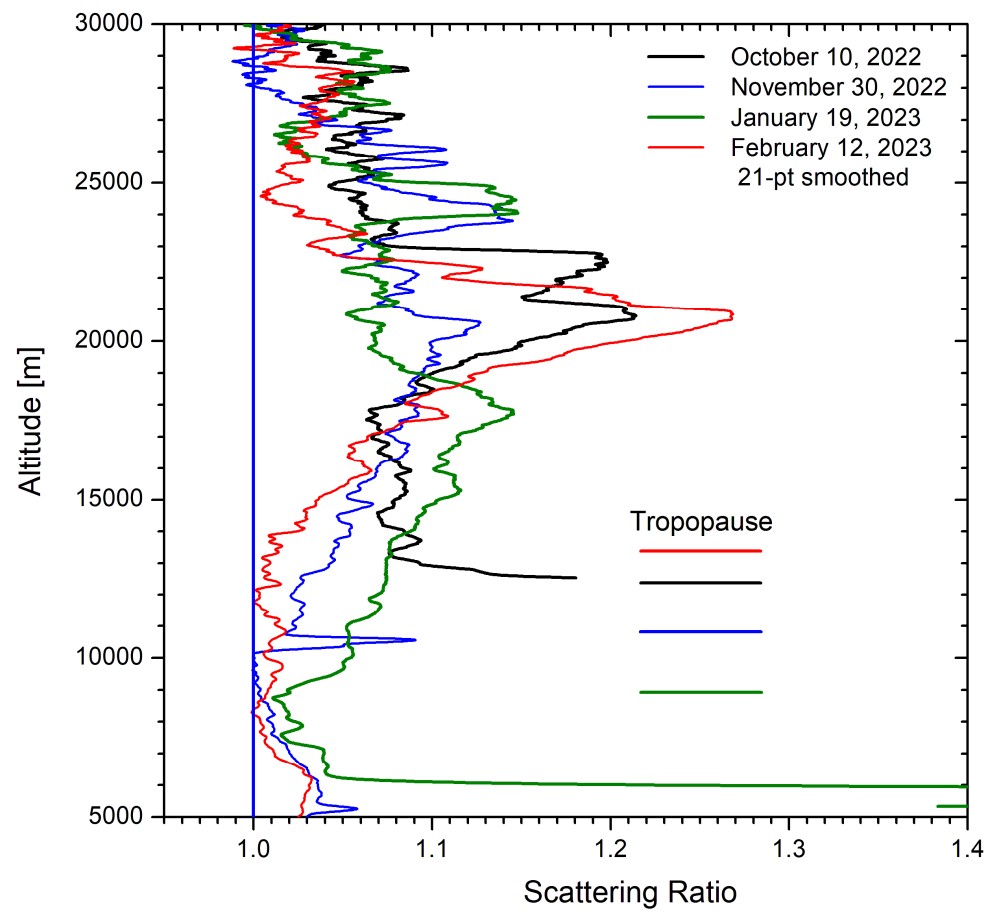

**Fig. 17.** Four selected profiles of aerosol scattering ratios between October 2022 and February 2023; the data are smoothed with ±10-bin gliding arithmetic averages. The tropopause altitudes are marked in the colours of the respective scattering ratios.




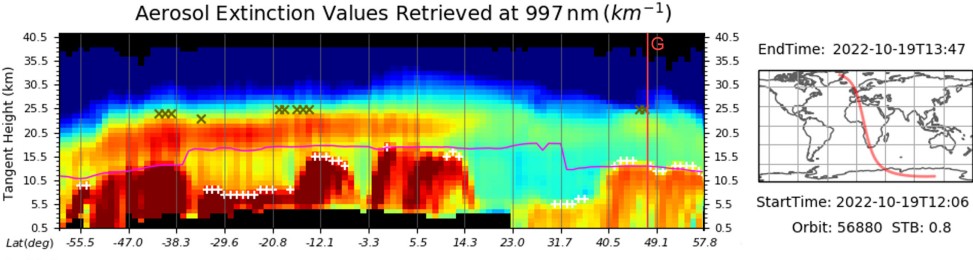

**Fig. 18.** Section of an OMPS curtain of the aerosol extinction coefficient along an orbit passing not far from Garmisch-Partenkirchen depicted in the right panel (19 October 2022); the elevated stratospheric aerosol caused by the Tonga eruption is located between roughly 56º S and 15º N, dark crosses marking pronounced aerosol layers. Slightly elevated aerosol is also seen around 47º N where also a few dark crosses are visible at 25 km. The violet line corresponds to the tropopause. The vertical red line visualizes the latitude of Garmisch-Partenkirchen (G).

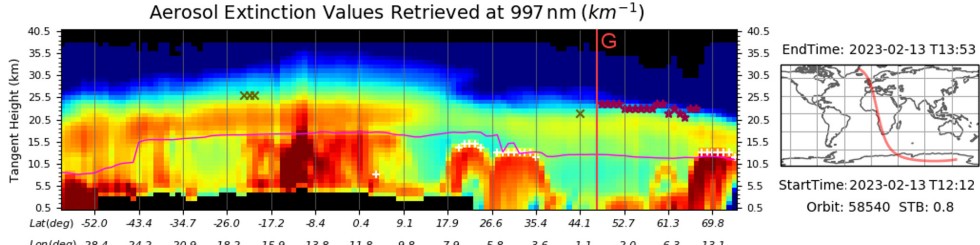

**Fig. 19.** Section of an OMPS curtain of the aerosol extinction coefficient along an orbit passing to the west of Garmisch-Partenkirchen depicted in the right panel (13 February 2023); dark crosses marking pronounced aerosol layers, red asterisks polar stratospheric clouds as determined by the OMPS algorithm. Slightly elevated aerosol is also seen around 47º N where also a few dark crosses are visible at 25 km. The violet line corresponds to the tropopause. The vertical red line visualizes the latitude of Garmisch-Partenkirchen (G).