# Peer review of "Measurement report: Violent biomass burning and volcanic eruptions: a new period of elevated stratospheric aerosol over Central Europe (2017 to 2023) in a long series of observations"

_EGUsphere, 2023_

## Author Comment (AC1)

**Reply to the three Comments on egusphere-2023-1781**

Thomas Trickl, November 30, 2023

The authors appreciate the very nice overall judgements on their paper and the valuable suggestions made. Here, we present our replies on the various comments received. The reviewers´ lines are given in italics. In addition to the modifications we decided to include Dr. Wolfgang Steinbrecht in the list of authors because of his substantial work to prepare the climatological ozone profiles.

**CC1: Comment on egusphere-2023-1781, Albert Ansmann, 18 Oct 2023**

*Dear Thomas and colleagues!*

*I read the manuscript with great interest. It is amazing to see this almost 50 year long time series of stratospheric aerosol observations, from 1976 to 2023.*

*The title of the manuscript points to the recent years (2017-2023) with a strong contribution of wildfire smoke to the aerosol in the stratosphere.*

*Because we from TROPOS worked a lot on this topic (and published several papers on wildfire smoke) I was forced to write a short comment.*

*Line 36: Yes, Mike Fromm pushed the smoke research forward (six references), but one could add even papers of Ohneiser et al. (ACP, 2022) on long term observations of Australian smoke (2020-2021) and Baars et al. (ACP) on Europe-wide Canadian smoke observations in 2017-2018.*

Ohneiser et al. are now cited here. Baars et al. and other papers on the 2017 fires are already cited in lines 59-60. Ohneiser et al., was not published when the Introduction was written which explains its omission!

*line 42: The decay time should be well defined in the paper. The e-folding decay time was 13-15 months in the case of Pinatubo over Northern Germany (Ansmann, JAS, 1997) and 1.14 and 1.37 years over Japan and New Zealand (Nagai et al., SOLA, 2010, https://doi.org/10.2151/sola.2010-018).*

Added

*line 60: Baars et al. (2019)*

Corrected

*line 77: Are you sure that the Colorado fires were responsible for the smoke pollution in 2020? There were also long-lasting record breaking Californian fires during the 2020 summer half year, as described in Hu et al. (2022) and Mamouri et al. (ACPD, 2023).*

I agree as to summer. However, in this case the plumes arrived over our site astonishingly late. This is described in the corresponding part of the "Results" section. We forgot to mention our HYSPLIT ensemble trajectories that verify the possibility of direct advection (indicated by the narrow aerosol structures). This is now done. Two references were added in the corresponding subsection of "Results".

*line 261: ...to more than 20 km (e.g., Baars et al. 2019, Torres et al., JGR, 2020 10.1029/2020JD032579).*

Added

*line 285: Meanwhile it seems to be well established that pyroCb lofting followed by significant self lofting (as long as the smoke optical depth is high, during the first days after injection) enables the smoke to reach heights of more than 20 km or even more than 30 km (Torres et al, 2020, Khaykin et al., 2018, 2020, Ohneiser et al., 2023). There are many more papers on this topic in the recent literature.*

This aspect is discussed! We add Torres et al., and Ohneiser et al.

*line 297: For the same reason (impact of these lofting processes), the main message regarding the use and applicability of HYSPLIT trajectories should be that such a trajectory analysis can only describe the smoke transport when the lofting processes are (widely) over, when the smoke reached already heights in the lower stratosphere. The impact of pyroCbs and self lofting are not considered in these trajectory simulations.*

As mentioned: Self-lofting is of course included in the statements in lines 285-287 and 304-305. We are, nevertheless, astonished that the HYSPLIT results are that good.

*line304: hot pyroCb is misleading..... after dissolution of the 'cold' pyroCb-related cirrus umbrella, smoke absorption and self lofting takes over.*

I do not like the expression "self-lofting" since this process is based on the absorption of energy; we discussed this matter and decided to removed "hot" since "complex meteorology" is enough.

*line 332: Also the lidar at Hohenpeissenberg detected the aerosol in the free troposphere on 2 October 2017.*

Thank you for this information! However, I think there are even more stations where the aerosol was observed. In our case, we take profit from colocation of the different sounding channels.

*lines 343-345: In this context one should provide the reference to the paper in which the Leipzig observation are presented and discussed: Ansmann, Frontiers in Environ. Sci., 2021, doi: 10.3389/fenvs.2021.769852.*

Added

*line 352: You assume that the smoke originated from strong Colorado fires? The articles of Hu et al., ACP, 2022, doi.org/10.5194/acp-22-5399-2022, Michailidis et al., ACP, 2023, doi.org/10.5194/acp-23-1919-2023, and Mamouri et al, ACPD, 2023 report smoke from record-breaking Californian fires. Maybe check again to be sure that Colorado fires were dominating.*

This possibility was already discussed (see above). We add a sentence about tropospheric observations of smoke plumes from North America over the Mediterranean region by the end of October, citing Michailidis et al. and Mamouri et al.

*line 403: To repeat, pyroCb contributes to the lofting of smoke, but also subsequently occurring self lofting of smoke.*

We copy a remark on the "complex meteorology of a pyroCb" from the 2017 section.

*line 414: You mean Boone et al. (2022)!*

Thank you! Corrected

*line 422: Do you think you observed volcanic ash or volcanic sulfate? Please discuss a bit.*

This already done on lines 414-416!

*line 550: Stenchikov et al. (2022) describes a similar scenario for Pinatubo aerosol as I tried to describe it for the smoke above. Pinatubo ash (plus SO2) was lofted (injected) up to about 17 km and then self lofting (as a result of absorption of solar radiation by ash particles) lofted the entire volcanic pollution higher up to 25-30km height (within a few days...). Then self lofting stopped because the high optical depth of ash disappeared by dilution and by sedimentation of ash particles. Then the sulfate layer formed in these lofted plumes with maximum heights around 30 km.*

I could not find (Stenchikov et al., 2022). The Pinatubo topic is discussed in 2021 by this team and we, thus, assume "2022" is a misprint. In addition, I found (Ukhov et al., 2023), but this study is limited to the early phase of the plume propagation..

*Figure 2: I do not understand! Please state clearly what you did! I speculate you applied the ozone correction to the lidar signals first and afterwards you applied the Klett method to the ozone-corrected signal profiles?*

This is correct. We add "raw" to backscatter signals in line 199.

*Figure 7: Californian or Colorado smoke?*

Discussed in the text!

*Figure 10: Such a profile structure with aerosol within the upper troposphere as well as in the lower stratosphere on 22 August 2019 (red profile) was also found over Leipzig on 14 August 2019 (Ansmann et al., Frontieres, 2021) and assigned as smoke layer because of the high 532 nm lidar ratios (unusually high compared to volcanic sulfate lidar ratios published by Horst Jaeger). The fact that the aerosol layer was not clearly above the tropopause seems to be a hint that this is some kind of an undefined aerosol layer, partly containing smoke in the lower part and volcanic sulfate in the upper part.*

Thank you for this hint! I added this reference in the text in two subsections. Please, note that in the figure caption no interpretation is given. This is done in the text.

*All in all a nice...., or better an excellent work!*

We really appreciate this applause.

**Review RC1: Comment on egusphere-2023-1781, Juan Carlos Antuna-Marrero, 20 Oct 2023**

*The manuscript reports the characterization of the stratospheric aerosols layer enhancements over Central Europe produced by explosive volcanic eruptions and severe smoke from big fires. To that end, lidar observations between 2017 and 2023 at Garmisch-Partenkirchen, one of the longest stratospheric aerosols lidar records is used. It has been extensively used for research up to the present as the references show.*

*The manuscript's scientific significance and scientific quality are excellent. The presentation is also excellent.*

*The description of the main features of the lidar instrumentation evolution and the most relevant aspects of the retrieved signal processing are included appropriately. Also, the discussion in the Annex of the contributions to the aerosol backscatter uncertainties is very*

*important. It addresses relevant scientific questions for ACP Journal, in particular the characterization of the vertical and temporal evolution of the stratospheric aerosol optical properties originated from severe smoke from big fires and explosive volcanic eruption. All those facts warrantees the traceability of the results. The time frame are the last 6 years in which the frequency of those severe smoke from big fires has increased without a plausible explanation so far.*

*The scientific results and conclusions are presented in a clear, concise, and well structures way with an appropriate number of figures and tables. It gives appropriate credit to similar research and clearly indicates its own contribution. The manuscript title is appropriated, reflecting its content. Similarly, the abstract provides a concise and complete summary.*

*The references area appropriated in quality and number as well as the supplementary material.*

*I have some comments and suggestions listed below.*

*Comments:*

*In the section "Spectacular pyro-cumulonimbus in British Columbia on 30 June 2021", the Figure 8 shows the aerosol backscatter uncertainty profile allowing to appreciate its reasonable magnitude respect to the aerosol backscatter. However, that is not possible for the two peaks going beyond the border of the figure. I suggest the authors include a brief description of the relative magnitudes of the uncertainties at the peaks in comparison with the relative magnitudes of the uncertainties from the rest of the profile.*

The relative uncertainty at 15.6 km is specified in the data file as 18 %. I added this information to the caption.

*Line 26: In the sentence: Ground-based lidar with its good vertical resolution became an important tool "almost" right from the beginning.... I suggest eliminating the word "almost" and add 2 more references: Fiocco, G., and G. Grams, 1964 and Elterman et al., 1973.*

I cited Fiocco and Grams in our 2013 paper in ACP. Here, we focus just on the long-term operations and I added "of long-term sounding".

*The first series of stratospheric aerosol profiles from lidar (Fiocco, G., and G. Grams, 1964, https://doi.org/10.1175/1520-0469(1964)021<0323:OOTALA>2.0.CO;2 ) and from searchlight (Elterman, et al., 1964, https://doi.org/10.1175/1520-0469(1964)021<0457:AAOWSP>2.0.CO;2) were extensively used for research on stratospheric aerosol from the middle of the sixties (after the Mt Agung eruption) until around the end of the seventies. They contributed to characterize the volcanic stratospheric aerosols vertical profiles evolution and the further advance of the scientific knowledge in the sixties and early seventies of the XX century. That is evidenced by the multiple articles referencing those results, including the references to Fiocco and Elterman provided in the two papers already cited in that sentence: McCormick et al., 1978; Simonich and Clemesha, 1997.*

*Elterman himself carried out research using Fiocco´s and his own datasets, (ex. Elterman et al., 1973 https://doi.org/10.1364/AO.12.000330; Elterman,1976 https://opg.optica.org/ao/abstract.cfm?URI=ao-15-5-1113).*

Thank you for passing this information! However, I could not find a hint on long-term operation in these publications.

*Line 342: For clarity I suggest to change the sentence: " The high lidar ratio suggested that observed at stratospheric aerosol at high latitudes were caused by import of fire smoke from Siberia...." By "The observed stratospheric aerosol high lidar ratio suggest its origin was the fire smoke from Siberia"*

The sentence is, still, not really good. I change it to "The high lidar ratio indicated that the stratospheric aerosol observed at high latitudes was smoke. The air mass could be traced back to fires in Siberia".

*Line 487: Is this sentence referring to figure 18.? There is not an October 19 profile in the figure, unless the one from that month was labeled as October 10, before the date cited in the sentence.*

Thank you for pointing this out. There was no measurement on October 10. I changed the date in Fig. 17.

*Line 1020: The link http://www.trickl.de/Rayleigh.pdf does not work.*

Indeed, it has not worked since my personal web site has not yet been fully revised after a server crash. Several files cited in previous publications are missing as well. The revision will be completed in the second week of December.

*Figure 3: 4th line in the caption. The acronym "PSC" is not defined in the text, been used commonly for Polar Stratospheric Clouds. The Quebec (1991) and Chilshom (2001) fires mentioned do not appear to be related to PSC events. Please correct or clarify it.*

This was done in our 2013 paper in ACP. Obviously, the acronym remained in the first version of the figure. The sentence is now removed.

*Figures 7: Add the wavelength the aerosol backscatter coefficient was measured.*

Added

*Figure 12: Briefly describe in the text the criteria to determine the upper and secondary boundaries of the top aerosol layer or provide a reference where it is described. Clarify the term main layer. Include in the caption the wavelength of the SCR. I suggest using the right axes scale for labeling the SCR magnitudes.*

The layer boundaries were not determined with a sophisticated mathematical procedure. They were estimated from approximate zero transitions. The term "main layer" is explained in the caption!

**Review RC2: Comment on egusphere-2023-1781, Sergey Khaykin, 30 Oct 2023**

*Review of Trickl et al., "Measurement report: Violent biomass burning and volcanic eruptions: a new period of elevated stratospheric aerosol over Central Europe (2017 to 2023) in a long series of observations".*

*The manuscript by Dr. Thomas Trickl and coauthors presents the updated time series of stratospheric integrated backscatter from the 50-yr long lidar profiling record at Garmisch-Partenkirchen and focuses on the lidar observations of stratospheric aerosol layers produced by intense wildfires and volcanic eruptions since 2017.*

*The authors should be congratulated for an impressive effort to produce such a long-term and high-quality observation record of stratospheric aerosol loading as well as for their rigorous approach to data quality and error budget assessement.*

*The manuscript is well structured, the experimental setup and the quality aspects are described in a comprehensive way, whereas an appropriate credit is given to the related literature. Overall, the study represents a valuable contribution to the stratospheric aerosol problematics from the observational perspective.*

*In my opinion, the study could potentially be a good match for the "Research article" category had it included a more profound analysis of the locally observed features (beyond attribution to a specific event using simple trajectory modeling), or had it provided new information on the aerosol optical properties from various sources or otherwise re-evaluated different transport pathways and their timescales. While there are several novel aspects invoked (e.g. BDC-driven transport of aerosols, prolonged stratospheric aerosol decay etc.), the related interpretation largely rests upon general considerations without appropriate support from other data sources, in particular the global observations.*

*The high-quality local observations presented in the paper will surely motivate further research on the topic using global observations and modeling experiments. Therefore, in no way the "Measurement report" category could reduce the scientific impact of this paper. I recommend it for publication in ACP after a few minor revisions as suggested below.*

*Specific remarks*

*l.336-337. It is indeed very interesting (and also puzzling) why the backscatter enhancements are restricted to the lower part of the ozone enhancements. Could it be linked to aerosol sedimentation within the intruded airmass? I would be extremely curious to see the time curtains of backscatter (or just the range corrected signal) together with ozone curtain if this is not too much work.*

I do not have any interpretation. Trickl et al. (2016) discuss a similar case. In their Fig. 12 the aerosol layer is located in the upper part of the intrusion layer. I added a sentence about this.

*l.379. Is the upper boundary of enhanced aerosol layer at 19.5 km attributed to wildfire smoke? To my knowledge, neither OMPS-LP nor CALIOP have reported aerosol layers this high during that time. If it were a smoke layer at this altitude, one should expect a significant self-lofting of highly-concentrated plumes prior to this lidar measurement, which would be readily detected by satellite sensors.*

The analysis of the observation is discussed in the following!

*l.435-440. While the wildfire smoke could, to some extent, contribute to the SA load perturbation after Summer 2019, I believe that the prolonged decay is largely due to the self-lofting of Raikoke sulfur-coated ashes as argued upon in this study https://www.nature.com/articles/s41598-022-27021-0 The self-lofting of aerosols is expected to prolong the aerosol removal from the stratosphere through gravitational settling.*

I added a sentence on this.

*l.439-440. I am unconvinced that the layers above 30 km could be attributed to Ulawun sulfates, please see my further remark on that matter.*

We also are not completely sure as pointed out.

*l.498-509. The detection of hydrated layers from Hunga using RS and lidar in the northern midlatitudes is certainly an important result, however this should be supported by the respective graphical material.*

I agree. However, we decided not to present that material (radiosondes, Raman lidar) because of the high uncertainty causing quite different values from case to case. Of course, I would love to show the lidar $H_2O$ data. The sonde values also scatter a lot, but are clearly above the typical stratospheric background level of 0 to 1 % RH. We plan to make comparisons.

*l.548-552. In terms of the amount of injected material and injection altitude, the Ulawun eruption is nowhere near that of the 1991 Pinatubo eruption. Thus, appealing to modeling results by Stenchikov et al. for a major eruption appears to me unjustified.*

Thank you for pointing this out. However, why should we not mention this? In any case, the layer is extremely weak. Instead, independently mentioning another possibility such as the Australian fire (see below) should be preferred. However, the Australian fire occurred rather late. We discuss this issue.

*l.557-558. Indeed, the absence of aerosols beyond 32 km after Pinatubo puts in doubt the attribution of layers above 30 km to the eruption of Ulawun. During the BDC-driven transport to midlatitude, the Ulawun sulfates would be exposed to warm temperatures and evaporate. This explains the absence of volcanic aerosols transported from the tropics at such altitudes. A much more plausible source of the high-altitude aerosols observed by the GP lidar, in my opinion, is the Australian ''Black Summer" PyroCb outbreak that generated a smoke-charged vortex rising up to 35 km whilst travelling towards the tropics. The fine smoke particles sediment slower than sulfuric acid droplets, which further corroborates this hypothesis. Another potential source could be the meteoritic dust. While an accurate attribution of the observed feature would require a careful analysis of various satellite data and transport modeling, I suggest elaborating this discussion a bit to consider various potential sources.*

Thank you for this hint. I added a sentence on this following line 552. We believe that this fire event took place too late to serve as a source of our observations although the upper boundary of our layer rises with time.

*l.582. One should also mention particle sedimentation (e.g. Kremser et al., 2016), which does not fall under strat-trop transport category.*

Added

*l.583. As far as the dilution and advection of clean air masses are concerned, it might be worth referring here to Vernier et al., ACP, 2011 and Khaykin et al., ACP, 2017 pointing out the poleward transport of convectively-cleansed air from the tropical tropopause region.*

Thank you: Added

**RC3: Comment on egusphere-2023-1781, Anonymous Referee #3, 06 Nov 2023**

*General comments:*

*This paper documents one of the longest lidar records of stratospheric aerosols in existence. Changes in the instrumentation and analysis are followed through the record, and their implications for the data are assessed. Numerous volcanic eruptions and wild fires*

*impacting this mid-latitude site in Germany are identified from sources in both the Northern and Southern hemispheres. This is a useful compilation for those wondering if a specific event impacted this region of the earth at a certain time. The absolute differences between events can also be reliably compared.*

*The paper is well written and only minor changes are suggested.*

*Specific comments:*

*Abstract: no comments*

*Line 45: Do you mean lifetime or 1/e folding time? I usually see about 1 year for these?*

Modified (also suggested by Dr. Ansmann).

*Line 129: "expensive additional laser photons", do you mean you can use a smaller laser?*

Indeed, we do use a smaller laser, a laser with lower 532-nm pulse energy. However, this is a general statement.

*Line 153: The field campaign didn't involve lidar right? It isn't portable, but was controlled from the United States?*

Of course, this field campaign did not involve the UFS aerosol lidar. A mobile $H_2O$ DIAL was used. I no write "routine measurements at UFS".

*Line 165: "Afterwards, an extended-Klett (Klett, 1985) program, originally developed and very successfully quality assured for aerosol retrievals within EARLINET, is used"*

"has been used" added

*Line 193: "The counting noise level in the raw data descends with altitude" Do you mean decreases with altitude?*

Yes: Changed

*Line 283: "(2018 " missing ")"*

"2017" added

*Line 330: "...Northern Canada tree..."?*

Thank you: three!

*Line 430: "The temporary minimum of the integrated backscatter coefficient in September, 2019 is caused ..."?*

Changed

*Line 494: "Not always stars exist ..."?*

New: Around the latitude of UFS (47.5º N) no cross exists anyway

*Line 505: What do you mean by depolarized particles?*

I am sorry: depolarizing

*Figure 4: caption, "...Munich tropopause altitudes from the Munich radiosonde ..."*

New: The corrected tropopause altitudes from the Munich radiosonde

*Figures 16, 18, 19: Can you show the color scale? If all three figures use the same scale you could show the scale for just figure 16 and refer to that in 18 and 19.*

In Fig. 16 no unit is given. Nevertheless, in all three cases the color-code bar is now added.